# Selective replication and vertical transmission of Ebola virus in experimentally infected Angolan free-tailed bats

S. A. Riesle-Sbarbaro [1], G. Wibbelt [2], A. Düx[1,5], V. Kouakou[3], M. Bokelmann[1], K. Hansen-Kant[1], N. Kirchoff[1], M. Laue [1], N. Kromarek[1], A. Lander[1], U. Vogel[1], A. Wahlbrink[1], D. M. Wozniak [1,6], D. P. Scott[4], J. B. Prescott [1], L. Schaade [1], E. Couacy-Hymann[3,7] & A. Kurth [1] ✉

The natural reservoir of Ebola virus (EBOV), agent of a zoonosis burdening several African countries, remains unidentified, albeit evidence points towards bats. In contrast, the ecology of the related Marburg virus is much better understood; with experimental infections of bats being instrumental for understanding reservoir-pathogen interactions. Experiments have focused on elucidating reservoir competence, infection kinetics and specifically horizontal transmission, although, vertical transmission plays a key role in many viral enzootic cycles. Herein, we investigate the permissiveness of Angolan free-tailed bats (AFBs), known to harbour Bombali virus, to other filoviruses: Ebola, Marburg, Taï Forest and Reston viruses. We demonstrate that only the bats inoculated with EBOV show high and disseminated viral replication and infectious virus shedding, without clinical disease, while the other filoviruses fail to establish productive infections. Notably, we evidence placental-specific tissue tropism and a unique ability of EBOV to traverse the placenta, infect and persist in foetal tissues of AFBs, which results in distinct genetic signatures of adaptive evolution. These findings not only demonstrate plausible routes of horizontal and vertical transmission in these bats, which are expectant of reservoir hosts, but may also reveal an ancillary transmission mechanism, potentially required for the maintenance of EBOV in small reservoir populations.

Forty-six years after the first Ebola virus disease (EVD) outbreak, a sporadic yet devastating zoonotic disease caused by Ebola virus (EBOV; *Orthoebolavirus zairense*), there is no convincing evidence identifying a natural reservoir host (R$_h$), nor the mechanisms of virus circulation in nature or spillover to humans. The burden of filovirus outbreaks to societies is transversal, with the aftermath taking a toll on public health[1], economy[2] and wildlife conservation[3].

The epidemiological unpredictability of filovirus outbreaks[4] and the clinical severity of the disease has prompted interdisciplinary research efforts, including modelling of filovirus pathology and maintenance in potential R$_h$[5–7]. Models of EBOV persistence in still-unidentified bat populations have been informed by extrapolating pathogen-R$_h$ interactions[8] of the related Marburg virus (MARV; *Orthomarburgvirus marburgense*) and Egyptian rousette bats (ERB; *Rousettus aegyptiacus*,

[1]Center for Biological Threats and Special Pathogens, Robert Koch Institute, Berlin, Germany. [2]Leibniz Institute for Zoo and Wildlife Research, Berlin, Germany. [3]LANADA, Laboratoire National d'Appui au Développement Agricole, Bingerville, Côte d'Ivoire. [4]Rocky Mountain Laboratories, National Institutes of Health, Hamilton, MT, USA. [5]Present address: Helmholtz Institute for One Health, Greifswald, Germany. [6]Present address: Bernhard-Nocht-Institute for Tropical Medicine, Hamburg, Germany. [7]Present address: Centre National de Recherches Agronomiques, LIRED, Abidjan, Côte d'Ivoire. ✉e-mail: kurtha@rki.de

E. Geoffroy, 1810), a $R_h$[9]. Wildlife surveillance, aimed at discovering $R_h$ of EBOV[10], have reported several species of frugivorous and insectivorous bats harbouring antibodies reactive to EBOV, Reston virus (RESTV; *Orthoebolavirus restonense*)[11] or other orthoebolaviral antigens[12]. However, undiscovered filoviruses or cross-reactive viruses in these bats, e.g. the recently discovered Bombali virus (BOMV; *Orthoebolavirus bombaliense*)[13], might have contributed to these seroprevalence data, within and outside of Africa.

Circumstantial evidence has associated the insectivorous Angolan free-tailed bat (AFB; *Mops condylurus, A. Smith, 1833*), a $R_h$ of the orthoebolavirus BOMV[14], as a spillover species triggering the 2013–2016 western Africa EVD outbreak[15]. Furthermore, bats that have biannual and synchronous breeding, as AFBs, are predicted to be suitable hosts to maintain EBOV in nature[8]. However, research efforts have been biased towards pteropodid fruit bats[16] due to the detection of very short fragments of EBOV RNA in three species of fruit bats[17] and the known ecology of MARV in ERBs, which was determined a MARV-$R_h$ only after years of comprehensive research and crucial evidence. A particularly relevant finding is the high reservoir competence that ERBs have demonstrated uniquely to MARV infection in vivo[5,18–20], but not following inoculation with other filoviruses[21]. Experimental infection of ERBs has consistently resulted in MARV replication in many tissues and a period of viremia. Further studies have demonstrated elicitation of antibody-mediated immunity[18,20], virus shedding[19] and horizontal transmission to conspecifics[5]. Vertical transmission, however, a key component of enzootic maintenance of some RNA viruses, including bat-borne[22,23], has not been investigated experimentally. Studies of transmission mechanisms of filoviruses in $R_h$, which could help conceptualize EBOV maintenance of enzootic cycles and drivers of spillover, are warranted. Only one pioneering experiment in 1996[7] showed that a human EBOV isolate (Kikwit variant) replicated in AFBs and a fruit bat, without morbidity. To date, these studies have not been repeated, verified or expanded. In this work, we inoculated 22 wild-caught AFBs with either EBOV ($n = 5$), MARV ($n = 5$), RESTV ($n = 6$) or Taï Forest virus (TAFV; *Orthoebolavirus taiense*; $n = 6$), of which 2 EBOV and 1 MARV -inoculated bats were gestating, allowing us to examine potential placental tropism and vertical transmission. We detected high and selective replication of EBOV in AFBs, in contrast to the other orthoebolaviruses or MARV that we used for inoculum. Additionally, we demonstrate the potential for horizontal transmission of EBOV and evidence the particular ability of EBOV to transmit vertically in AFBs. Thus, our study shows that AFBs may be relevant for the maintenance of enzootic cycles of EBOV and that are appropriate models to further investigate pathogen-$R_h$ interactions of this virus.

## Results

### Captivity, diet adaptation, and clinical observations

The inherent husbandry challenges that Swanepoel et al.[7] encountered while working with insectivorous bats decades ago (i.e. diet adaptation challenges resulting in starvation and death), were systematically addressed in our study. To adapt AFBs from their natural diet of flying insects[24] to a diet of mealworms (*Tenebrio molitor*, Linnaeus, 1758), we developed and optimized a feeding scheme (Fig. S1, Table 1). During a 10-week captivity period, the bats remained healthy, gained weight (Fig. S2) and three females conceived. This allowed us to measure the marked weight fluctuation of captive AFBs and to decrease inter-individual gut microbiome variability before experimental infection through standardized feeding[25]. Most importantly, it provided the opportunity to examine vertical transmission of filoviruses in AFBs. Bats were allocated into 4 virus cohorts (Fig. S3), which included 2 EBOV-inoculated dams, sampled at 5 (E03) or 10 (E05) dpi, and 1 MARV-inoculated dam sampled at 10 dpi (M05).

No mortality or clinical signs of disease were observed throughout the study (Fig. 1). The dams, which were synchronously at late-gestation, maintained a normal course of pregnancy without

**Table 1 | Successful virus isolation and titration from positive quantitative Reverse-Transcription PCR samples collected from adult filovirus-inoculated AFBs**

| TISSUE | EBOV 5 dpi PCR; | (VI) | EBOV 10 dpi PCR; | (VI) | MARV 5 dpi PCR; | (VI) | MARV 10 dpi PCR; | (VI) |
|---|---|---|---|---|---|---|---|---|
| Lymph Node | 3/3; | -- | 2/2; | (0/1) | 1/3; | (0/1) | 0/2; | -- |
| Uterus/ Placenta | 1/1; | (1/1) | 1/1; | (1/1) | | | 1/1; | (1/1) |
| Blood | 3/3; | (1/1) | 1/2; | -- | | | | |
| Spleen | 3/3; | (1/2) | 2/2; | -- | | | | |
| Colon | 3/3; | (1/1) | 1/2; | -- | | | | |
| Kidneys | 3/3; | (1/1) | 1/2; | -- | | | | |
| Bladder | 3/3; | (2/2) | 1/2; | -- | | | | |
| Testes | 2/2; | (1/1) | 0/1; | -- | | | | |
| Brain | 3/3; | (1/1) | 0/2; | -- | **RESTV** | | | |
| Salivary Gland | 2/3; | (0/2) | 1/2; | -- | | | | |
| Oral Swab | 3/3; | (1/2) | 0/2; | -- | **5 dpi** | | **10 dpi** | |
| Rectal Swab | 3/3; | (1/2) | 1/2; | -- | PCR; | (VI) | PCR; | (VI) |
| Spleen | 3/3; | (1/2) | 2/2; | -- | 2/3; | (0/1) | 0/3; | -- |
| Gallbladder | 1/1; | (1/1) | 2/2; | (1/1) | 1/3; | (0/1) | 2/3; | -- |
| Stomach | | | | | 2/3; | (0/1) | 0/3; | -- |
| Skin Inoculation | | | | | 3/3; | (1/2) | 0/3; | -- |
| Pooled faeces* | 3 dpi (+/+); 5 dpi (+/−); 10 dpi (+/−) | | | | | | | |
| Pooled urine* | 10 dpi (+/−) | | | | | | | |

PCR, polymerase chain reaction; VI, Virus isolation; dpi, days post inoculation.
**\*** Only dpi collection with attempted virus isolation shown: (PCR/VI). (+) Positive; (-) Negative; (--) Virus isolation not attempted: virus RNA loads below the cut-off.

foetal distress, confirming wellbeing and likely delivery. Daily weight change (Fig. 1a, c) did not deviate significantly between virus cohorts and control ($\chi^2 = 7.8$, df = 4, $p = 0.1$), or between periods of captivity, acclimation and infection in all experimental bats ($\chi^2 = 0.9$, df = 2, $p = 0.63$) or specifically in pregnant females (KW = 0.72, df = 2, $p = 0.7$). Bats maintained their weight over time; mean weight, from acclimation to experimental endpoint, was $31.8 \pm 4.6$ g (range: 20 - 48 g). However, broad daily weight fluctuations were often measured from captivity onwards; weight change ranged from -28% to +30% with peak deviations during transportation and the first day of acclimation, or for the control dam (NC) also after parturition (Fig. 1e). Likewise, similar to uninfected bats, blood chemistry showed no evidence of systemic clinical disease in any virus cohort (Fig. 1f). Overall, there was great heterogeneity of measured parameters among all groups: captives (wild-caught captive bats), controls (BSL4 controls) and virus-inoculated. We detected, however, a significant increase of total cholesterol (mg/dL) between cohorts (F = 3.4, $n = 44$, df = 5, $\eta^2 = 0.3$, $p = 0.01$), specifically captive ($n = 17$) compared to BSL4 control ($n = 9$) -bats ($p$-adjusted= 0.04), and a significant decrease of serum albumin (ALB mg/dL) between cohorts ($\chi^2 = 20.1$, $n = 42$, df = 5, eta2[H] = 0.4, $p = 0.001$), specifically EBOV-inoculated ($n = 3$) compared to captive ($n = 17$) -bats ($\chi^2 = 20.1$, df = 5, $p$-adjusted= 0.008) and MARV-inoculated ($n = 5$) compared to captive ($p$-adjusted= 0.03) -bats, neither significant compared to BSL4 controls.

### Selective and disseminated replication of EBOV in AFBs

To test the competence of AFBs as EBOV-$R_h$, we compared AFBs susceptibility to representative species of *Orthoebolavirus* and *Orthomarburgvirus* clades. Here, EBOV uniquely replicated in all AFBs inoculated with virus present in at least one sampled tissue of each

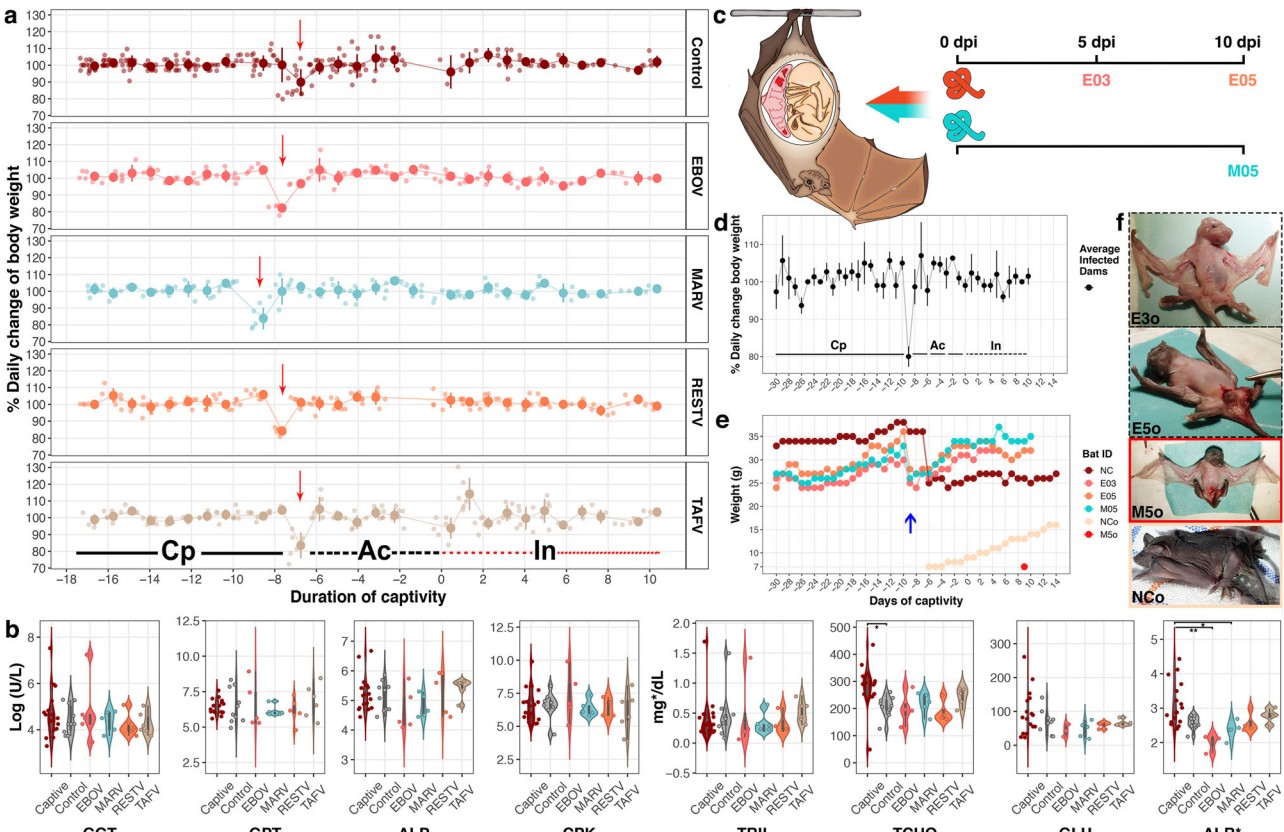

**Fig. 1 | Clinical progression of bats infected with filoviruses. a** Daily weight variation of bats including periods of captivity (Cp, $n = 37$) in Côte d'Ivoire, acclimation (Ac, $n = 33$) and infection (In, $n = 32$). In red, the dash line denotes the first In-period: 0–5 days post-inoculation (dpi), the dotted line denotes the second In-period (5–10 dpi). Circles show daily weight percentage change (DWPC) as mean values ± SD (error bars). Red arrows indicate transportation, highlighting the subsequent weight loss. No statistically significant difference was detected between groups or periods. **b** Blood chemistry of control and inoculated bats. Colour-coded cohorts show: Captive = wild-caught bats in Côte d'Ivoire ($n = 17$); Control = Captive and mock-inoculated bats kept in the BSL4 cages ($n = 9$); EBOV ($n = 5$), MARV ($n = 5$), RESTV ($n = 6$) and TAFV ($n = 6$) = bats inoculated with each virus, sampled at 5 and 10 dpi. Boxplots within violin plots show minimum, maximum, median and 25/75 percentile. Circles show individual bats. GGT, GPT, ALP, and CPK $y$-axis are log-transformed to improve visualization. ALB*= g/dL. Statistical significance, $p$ value < 0.05 (*) and <0.01 (**), measured using two-tailed tests: one-way ANOVA followed by Tukey HSD (TCHO, *p-adj* = 0.041) and Kruskal-Wallis rank sum test followed by Bonferroni multiple comparison adjustment in Dunn's test (ALB*, *p-adj* = 0.030 and *p-adj* = 0.008). **c** Sampling scheme of pregnant females inoculated with EBOV: sampled at 5 (E03) or 10 (E05) -dpi, and MARV: sampled at 10 dpi (M05). **d** DWPC of three infected pregnant females and one control (NC) bat, weighed during periods of Cp, Ac and In. DWPC (shown as mean values ± SD) did not differ significantly between periods. **e** Individual weight of pregnant females and offspring. The timeline is synchronized to day of BSL4-containment/arrival (blue arrow). The increasing weight of the negative control's offspring (NCo) is shown for reference. M05's foetus (M5o), weighed during necropsy, is shown in red. **f** Comparison of offspring physical development: between foetuses sampled at 5 (E3o) or 10 dpi (E5o and M5o) and 1-day old neonate (NCo). $n$ = biologically independent animals. Source data are provided as a Source Data file.

individual (Fig. 2a). Bat E04, which had the lowest EBOV-RNA loads (only spleen and stomach above the inoculum dose, Fig. S4), remained viremic at 10 dpi, and 5 of its tissues (including spleen) showed higher EBOV-RNA loads than blood, refuting erroneous virus in the blood contaminating tissues. Viremia was detected in all but one EBOV-inoculated bat (E05, 10 dpi): RNA loads (RNA copies/ml or g) in blood ranged from $1.8 \times 10^3$ to $2.0 \times 10^7$ at 5 dpi and $1.5 \times 10^3$ at 10 dpi (bat E04). EBOV tissue dissemination was evident at both time points and although no statistical analyses could be performed (10 dpi, $n = 2$), there was a clear distinction of viral loads between time points. Decreasing EBOV-RNA loads were detected from 5 to 10 dpi in all but one (cervical lymph node) tissue; bats euthanized at 5 dpi had up to 3 orders of magnitude higher EBOV-RNA loads than corresponding tissues sampled at 10 dpi. Aside from E03's individual EBOV-RNA loads detected in placenta ($1.5 \times 10^9$) and gallbladder ($1.8 \times 10^8$), the highest mean RNA loads in adult bats were detected in spleens ($M = 3.7 \times 10^8$, $n = 3$, Mdn = $9.3 \times 10^6$, IQR = $5.5 \times 10^8$) and stomachs ($M = 6.1 \times 10^7$, $n = 3$, Mdn= $1.4 \times 10^7$, IQR = $8.4 \times 10^7$) at 5 dpi. Similarly, placenta and spleens supported the highest EBOV-RNA loads at 10 dpi. Other tissues with detectable EBOV-RNA in all bats were liver, small intestine and cervical lymph nodes. Furthermore, from 16 representative sample-types (i.e. tissues with >$5 \times 10^5$ viral RNA loads, of foetal origin or secretions/excretions with >$2 \times 10^2$ RNA loads) infectious EBOV was isolated from 13 (Table 1), with infectious titres (TCID$_{50}$) and EBOV-RNA loads significantly correlated ($r_s = 0.8$ $p = 0.005$, n = 10).

In comparison, replication and dissemination of the other filoviruses was limited (Fig. 2a). Pairwise tissue comparison of mean virus-RNA loads was significantly different between EBOV and the other virus cohorts at 5 dpi (Q = 48, $n = 80$, df= 3, $p = 2.7 \times 10^{-10}$, W = 0.79). Viremia was detected only in one 10-dpi RESTV-inoculated bat ($5.1 \times 10^3$ copies/ml). RESTV-RNA was detected in 12 sample types of 23 tested per bat; the highest RESTV-RNA loads were measured in the skin area of virus inoculation sampled at 5 dpi (M = $9.2 \times 10^5$, $n = 3$, Mdn= $5.2 \times 10^5$, IQR = $1.1 \times 10^6$), with 3/3 RNA-positive bats. Isolation of infectious RESTV was attempted in bat R03's skin and 4 tissues of bat R02, including its skin (Table 1); the only tissue with successful RESTV re-isolation. MARV-RNA was detected in 8 different tissues, most abundantly in reproductive organs (3/5). Noteworthy, excluding placenta, the uterus of M05 had the highest MARV-RNA loads of all samples ($6.9 \times 10^5$ copies/g)

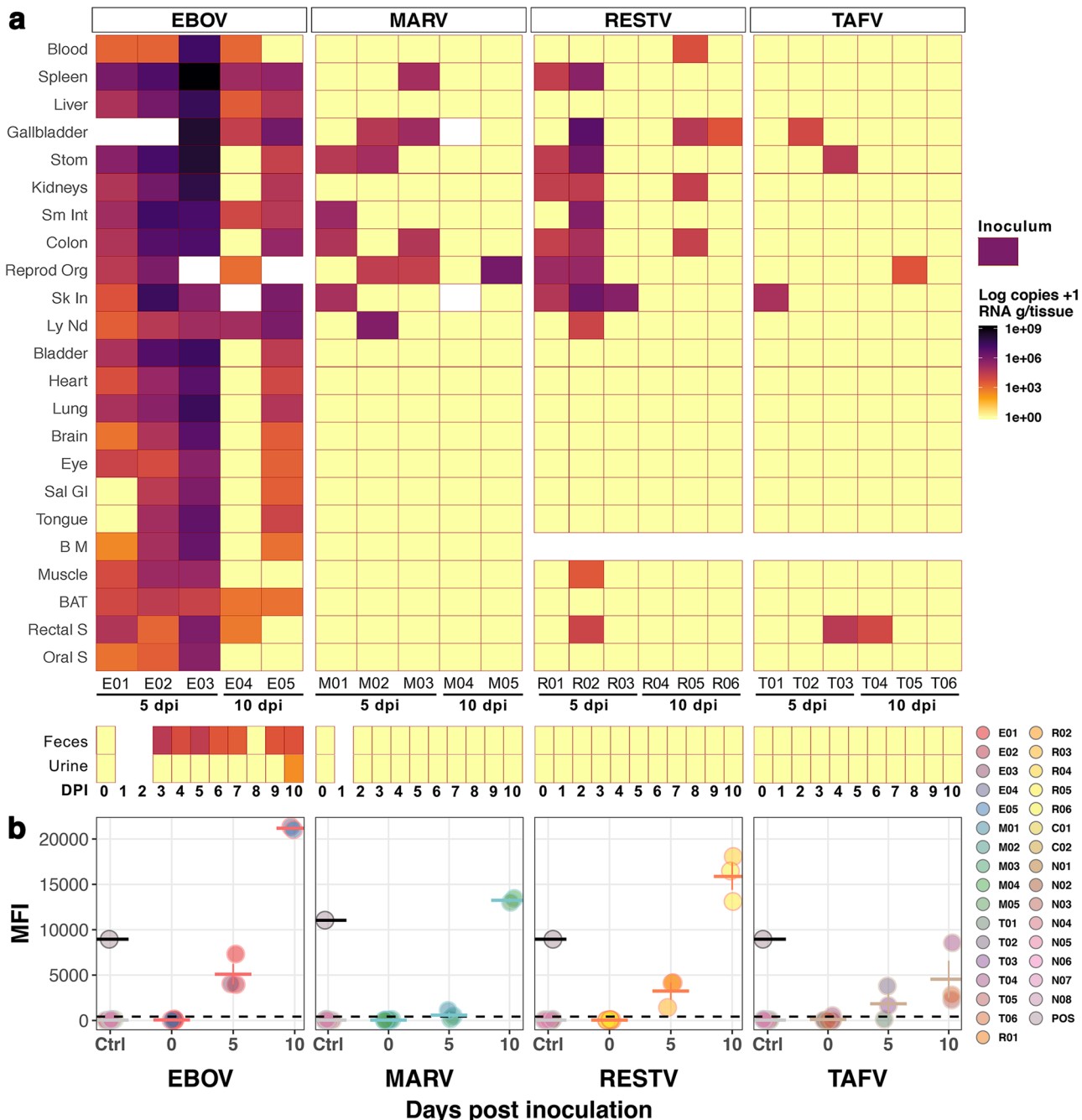

**Fig. 2 | Infection of Angolan free-tailed bats (AFBs) with filoviruses. a** Filovirus RNA copies/g of tissues or copies/ml of blood (rows) sampled from inoculated AFBs (columns) at day post-inoculation (dpi) 5 and 10. Bat identification numbers are noted below. Inoculum RNA load is shown as legend for reference. Blank spaces indicate unavailable samples (tissues, faeces and/or urine). Negative control bats are excluded Stom: Stomach; Sm Int: Small Intestine; Reprod Org: Testes or Uterus; Sk In: Inoculation point in Skin; Ly Nd: Lymph Node (cervical); Sal Gl: Salivary Gland; B M: Bone Marrow; BAT: Brown Adipose Tissue; S: Swab. **b** Serology (Luminex) of filovirus-inoculated (*n* = 22), positive (*n* = 2) and negative (*n* = 32) control AFBs. Circles show mean values ± SD (horizontal and error bars) of serum samples run in duplicate. Time point 0 shows pre-inoculation results, later re-analysed after euthanasia (5 and 10 dpi). EBOV and MARV positive controls from previously validated positive AFBs serum are shown as grey circles with black outline. Negative controls, including mock-inoculated (C01-C02) and bats kept in BSL4 caging (N01-N08) are shown in coloured circles with light-grey outline. The dashed line denotes the assay cut-off (mean value of negative controls + 3 SD). *n* = biologically independent animals. Source data are provided as a Source Data file.

and harboured infectious virus. Lymph nodes of bat M02 (5 dpi) had the second highest MARV-RNA loads (7.2 ×10⁵ copies/g), however, no infectious virus was isolatable. TAFV was the least abundant and disseminated, with low RNA loads detectable in only five sample types (Fig. S4), however, these included rectal mucosa of 5-dpi bat T03 and 10-dpi bat T04.

## Filovirus shedding in AFBs show EBOV's unique potential of horizontal transmission

Horizontal transmission of MARV has been inferred experimentally in a $R_h$[5] with the detection of infectious virus particles in tissues of the gastrointestinal and urinary systems, but it has been effectively evidenced by shedding of infectious MARV in daily bat secretions/

excretions (saliva, urine and faeces). Here, of all sampled secretions/excretions, we detected the highest RNA loads in the EBOV cohort; specifically, rectal mucosa sampled at 5 dpi (M = $9.1 \times 10^4$, $n = 3$, Mdn = $1.7 \times 10^4$, IQR = $1.3 \times 10^5$). Likewise, although EBOV- and TAFV-RNA was detected at 10 dpi (Fig. 2a), overall orthoebolavirus-RNA was most frequently and abundantly detected in rectal samples at 5 dpi. Amongst all filoviruses, however, only EBOV-RNA was detected in faeces and urine; in 7/8 faecal samples compared to in 1/8 urine samples (10 dpi, $3.2 \times 10^1$ copies/swab) tested. The highest RNA loads in faeces were detected at 3 ($2.1 \times 10^4$ copies/g) and 5 ($1.5 \times 10^4$ copies/g) dpi. In contrast, we found no evidence of MARV shedding, neither via secretions nor via excretions. Importantly, we detected infectious virus shed uniquely from EBOV-inoculated bats. Even though there was lower EBOV-RNA loads orally (M = $4.8 \times 10^4$, $n = 3$, Mdn = $6.9 \times 10^2$, IQR = $7.2 \times 10^4$) than rectally, infectious virus was successfully isolated from both oral and rectal mucosa of bat E03 at 5 dpi (Table 1). Furthering the potential of virus transmission between adult bats, infectious EBOV was evidenced in faeces collected at 3 dpi.

### Antibody-mediated immunity
ERBs have shown to develop humoral immunity to orthoebolaviruses, even though they do not support widespread infection[21]. Here, we similarly evidenced seroconversion of all inoculated AFBs, except for 1 TAFV and 2 MARV -inoculated bats sampled at 5 dpi (Fig. 2b). The average values of mean fluorescent intensities (MFI) per day in all virus cohorts reflected increasing antibody (Ab) titres from 5 to 10 dpi. However, this increase was not as dramatic in the TAFV cohort. For reference, we included previously validated positive AFB serum samples[26] as positive controls (POS) for orthoebolaviruses and MARV. Amongst all inoculation groups, EBOV elicited the highest Ab titres, with average MFIs at 10 dpi ($n = 2$) more than twofold greater than the POS. Average MFIs of 10-dpi RESTV-inoculated bats ($n = 3$) surpassed the POS by 1.7-fold and 10-dpi MARV-inoculated bats ($n = 2$) were just above the POS (1.17-fold). Differently, average MFIs of 10-dpi TAFV-inoculated bats ($n = 3$) were less than half the POS.

### Filovirus infection is not associated with gross pathology
During necropsy, abnormal macroscopic findings were rare (Supplementary Data 2). Noteworthy, several bats from all cohorts had abundant helminth infections in the digestive tract (Supplementary Data 3), and bat E01 and all MARV-inoculated bats, except bat M03 (5 dpi), presented with splenomegaly. Bats M04 and M05 (10 dpi) had congested spleens with irregular margins (Fig. S5).

### Association of filovirus antigen (Ag) with histopathological findings
All individuals had varying degrees of background pathology, as expected in wild-caught bats[14,27]. However, specifically supported by the immunohistochemistry (IHC) results, a few mild pathological changes could be attributed to infection, most of which were associated with EBOV at 5 dpi (Fig. 3a–c) and RESTV in skin sampled at 5 dpi. IHC of MARV and all 10-dpi-bats displayed no positive staining. TAFV-IHC was not performed. Detailed individual findings are included (Supplementary Data 3–4, supplementary text). Haematoxylin and eosin (HE) staining describes the general histopathological findings detected in all bats analysed (virus cohorts plus 3 control bats).

Almost all lungs presented blood vessel congestion by HE. Likewise, most bats had mild pathological background changes, e.g. neutrophilic infiltration of alveolar walls, pneumocyte hyperplasia and pulmonary arteriole hyperplasia. Yet, we detected virus-Ag only in bat E02 (Fig. 3a, c). As in a pioneering experiment[7], EBOV-Ag was detected in histiocytic cells of AFBs; EBOV-positive cells were limited to few lung lobuli containing multifocal macrophages in/next to alveolar walls (Fig. S6a–b).

In contrast to other animals, we observed that AFBs possess distinct round/ovoid islands of large histiocytic cells demarcating the lymphoid follicles (Fig. 3a, Fig. S6c). All animals had unremarkable to mildly reactive splenic lymphoid tissue. Still, EBOV-Ag was abundant in lympho-reticular tissues, with bat E03's containing: numerous IHC-positive histiocytic cells, predominantly surrounding the splenic lymphoid follicles, few IHC-positive macrophages in its tonsil lymphoid centre, and numerous in its thymus (Fig. S6d–f). Sparse IHC-positive cells were also detected in E03's bone marrow (Fig. S6g–h). Mediastinal lymph nodes of bats E02-E03 contained single distinctly EBOV-positive macrophages (Fig. S6i–j).

Livers presented with minor changes, restricted to periportal lymphoplasmacytic infiltration. We detected irregularly distributed degenerative to necrotic hepatocytes and distinct small necrotic foci with accumulations of macrophages, plasma cells and occasional lymphocytes (Fig. 3a). Bats E02-E03 presented evenly-disseminated EBOV-positive Kupffer cells (Fig. 3a, c, Fig. S6k–l), EBOV-positive necrotic foci and single hepatocytes (Fig. 3a, d).

Five (E01, R01-R03 and T01) of 21 bats sampled showed mild dermal/subcutaneous granulomatous inflammation (Fig. S6m–n); all but T01 presenting EBOV-positive macrophages.

### EBOV-Ag is associated with mild pathological changes in AFBs
To quantify the degree of pathological lesions associated with the presence of Ag, we selected bats with the highest Ag dissemination and abundance: E02 and E03. We estimated virus abundance as staining intensity within IHC-positive cell patches and a virus-associated pathology score (VAPS), according to virus-Ag co-localization with pathological changes. Generally, the selected tissues, which showed high viral RNA loads (Fig. 3b), presented varying degrees of Ag abundance (Fig. 3c). Nonetheless, the vast majority of the lesions observed were not associated with Ag presence (Fig. 4d). For example, the lympho-reticular tissues, some of which had high to very high viral RNA loads, presented with either low (spleen E02), medium (spleen, lymph node, thymus and bone marrow, E03) or high (lymph node E02) Ag abundance. Still, similar to the targeted liver pathology detected in ERBs infected with MARV[28], EBOV abundance was most frequently associated with hepatic lesions. Potentially due to the diverse tissues analysed herein, we observed pathological changes dependent on tissue-type rather than on viral load. Ag co-localization with areas of histopathological changes was detected in liver, lung and mesenteric tissues only; with the exception of necrotic foci in livers, all changes were associated with immune cell infiltration. In both analysed bats, IHC-positive Kupffer cells, monocytes and a few hepatocytes were detected. Bat E02, which had moderate Ag abundance, showed a few small focal necrotic foci associated with Ag. In contrast, a large but single necrotic focus in E03's liver was associated with EBOV-Ag presence. The lung had the highest IHC intensity-score in bat E02; EBOV-positive macrophages associated to the alveolar wall were detected within, and limited to, a few lobuli. In both bats, the mesenteric tissue was infiltrated with abundant macrophages that contained large amounts of Ag. Bat E02 had intensely stained macrophages adjacent to the jejunum and bat E03 had focally stained macrophages in the area of the mesentery between liver and kidney.

### Placental tropism of filoviruses and vertical transmission of EBOV in AFBs
In humans, both EBOV and MARV have shown to replicate extensively in placenta[29]. In AFBs, both filoviruses similarly presented targeted placental tropism; virus was isolated from placentas for all bats (Fig. 4). Importantly, EBOV uniquely traversed the placental barrier and infected foetuses. The highest EBOV-RNA loads in adults were detected in the placenta of E03 ($1.8 \times 10^9$ copies/g), which were only surpassed by EBOV-RNA loads detected in the liver of its foetus E3o ($3.5 \times 10^9$ copies/g). In fact, all the organs sampled from E3o ($n = 7$)

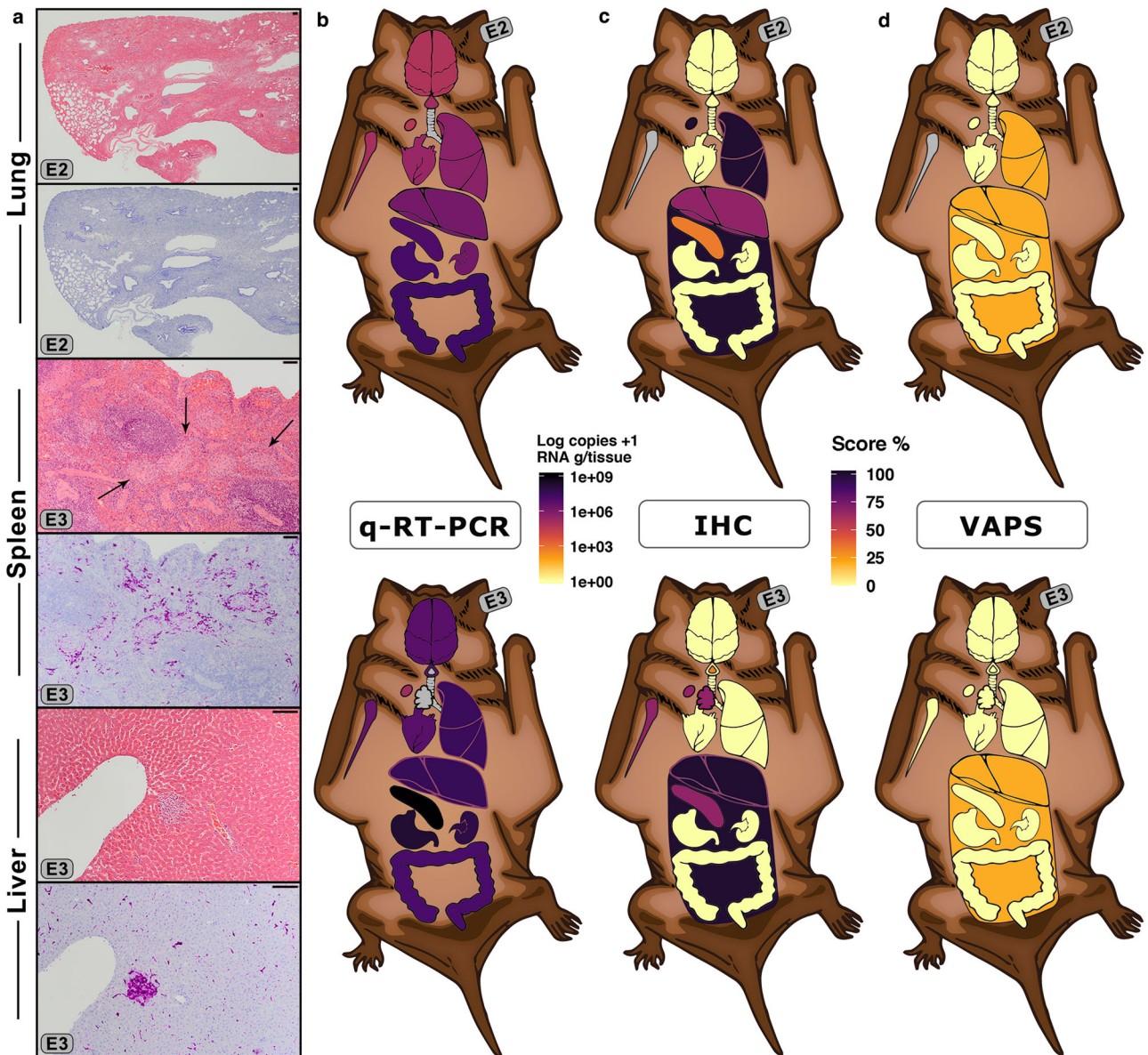

**Fig. 3 | Histology and immunohistochemistry (IHC) of filovirus-inoculated Angolan free-tailed bats (AFBs).** **a** Photomicrographs of EBOV-inoculated bats (images are representative of one experimental group and 3 independent animals). Tissues of E02 and E03, sampled at 5 dpi, stained with haematoxylin and eosin (HE) and IHC targeting EBOV VP40. Lung of bat E02 is shown stained with H&E and corresponding section with IHC. Spleen and liver sections of bat E03 are shown stained with H&E and IHC. Focal necrosis of the liver is shown by H&E staining, and co-localized EBOV antigen by IHC. Black arrows in spleen indicate islands of large histiocytic cells, physiologically found in this bat species. Scale bars, 100 μm. **b–d** EBOV dissemination and associated pathology in AFBs. **b** Heatmap representation of EBOV distribution in organs available for histology of bats E02 and E03, measured by RNA copies/g tissue (q-RT-PCR). **c** Abundance of EBOV antigen shown as a score of IHC staining intensity within cells (IHC). **d** Virus-associated pathology score (VAPS), defined by the severity of the cellular pathological changes directly associated with virus presence (co-localization with IHC). Unavailable organs are depicted in grey. Tongue is shown as a triangle above the trachea. Tonsil (shown as a triangle inside "tongue"), thymus and bone marrow were available only for bat E03 for IHC and VAPS. Lymph nodes (shown as ovals) were mesenteric (histology) and cervical (q-RT-PCR). Mesenteric adipose tissue from the abdominal cavity is shown as a rectangular background surrounding the abdominal organs. Min-max normalization of IHC and VAPS scores are shown as percentage. Source data are provided as a Source Data file.

had high viral RNA loads; the lowest ($1.1 \times 10^7$ copies/g) detected in thymus (Fig. 4a). EBOV RNA was also detected regularly within cell cytoplasm using RNA-in situ hybridization (RNA-ISH) probes (Fig. 4b, e) and infectious EBOV was isolated from several tissues of E3o (including liver), with correspondingly high $TCID_{50}$ titres (Fig. 4d). Similar to other tissues of the 10-dpi bat E05, its placenta had lower EBOV-RNA loads in comparison to the 5-dpi sampled E03 ($3.6 \times 10^6$), which was also reflected in tissues of its foetus (E5o). These results confirm the ability of EBOV to reach and traverse the placental barrier and replicate efficiently in foetuses of AFBs. Different than the focal staining seen in the liver of E03, EBOV was disseminated

throughout its placental endothelium, but strikingly so in E3o's tissues, particularly liver (Fig. 4e). This contrasting infection pattern is likely due to the immune tolerant environment, characteristic of the maternal-foetal interface. In contrast, MARV-RNA was not detected in any foetal tissue, even though the placenta of M05 had higher viral RNA loads than the placenta of E05 ($1.5 \times 10^8$ copies/g) and abundant intracytoplasmic MARV was detected within cells of its placental labyrinth (Fig. 4b, e).

Filovirus infection in pregnant humans (i.e. non-reservoir species), traverses the placenta and inevitably results in foetus (or newborn) infection and death[29]. Therefore, to robustly examine EBOV

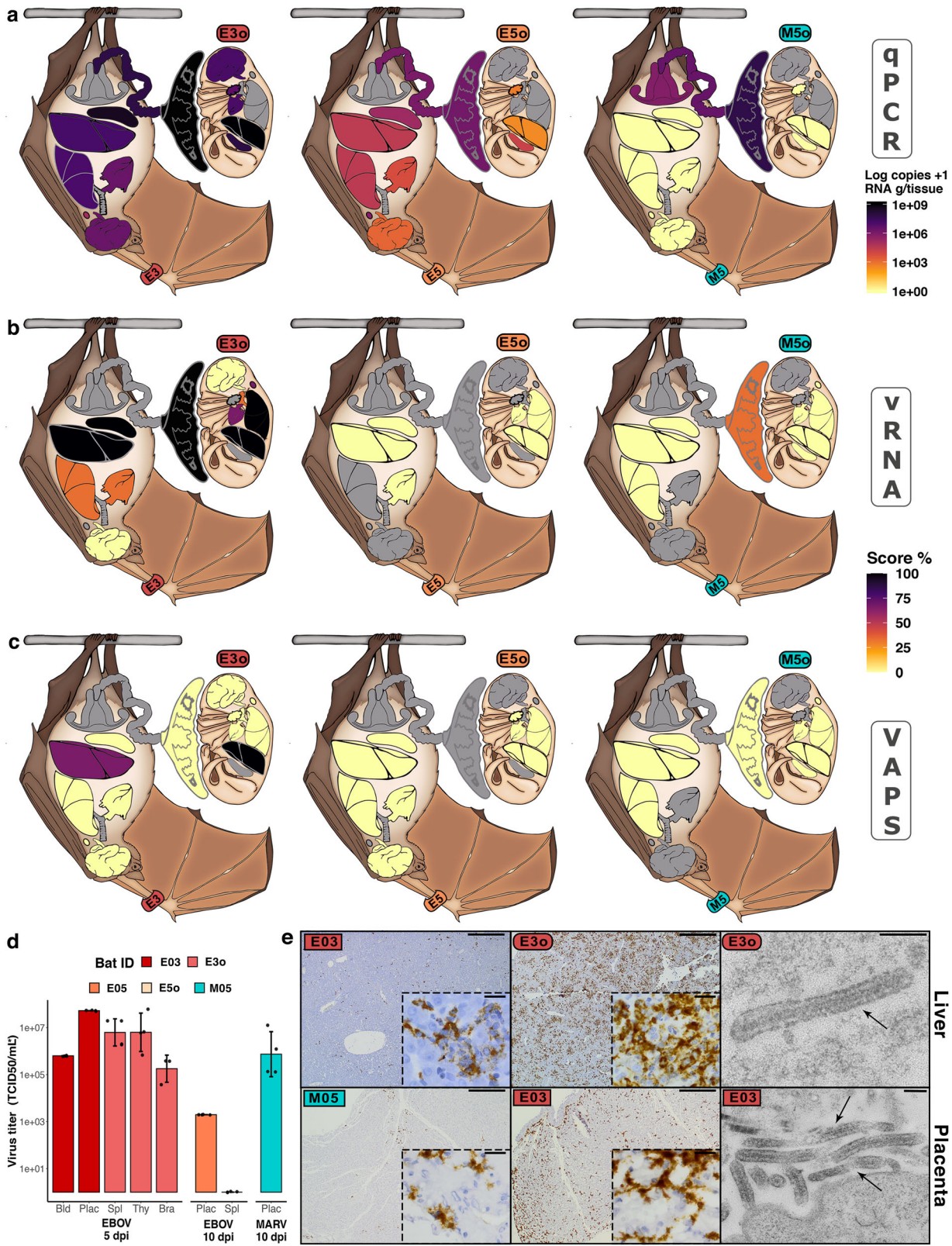

replication in AFB's foetus, we examined E03's placenta and E3o's liver by thin section electron microscopy (EM) for the presence of EBOV particles. Typical filamentous EBOV particles, found in the extracellular space between placental and liver cells (Fig. 4e) and presence of intracytoplasmic virus factories (Fig. S7) indicated replication. We also evaluated VAPS in foetus' livers; although no gross pathology was observed during the necropsy, the histopathology revealed necrosis in

E3o and immune cell infiltration in E5o (Fig. S8). Only in E3o we detected a tight association between the histopathological changes and EBOV presence (Fig. 4c).

## Distinct signatures of filovirus adaptive evolution

The slow evolutionary clock of orthomarburgviruses in R$_h$, characterized through time and between great geographical distance[30], is also

**Fig. 4 | Dissemination of filoviruses in experimentally inoculated pregnant Angolan free-tailed bats (AFBs). a–c** Representation of EBOV and MARV distribution in selected organs. **a** Heatmap representations of viral RNA measured by qRT-PCR (qPCR), shown as $\log_{10}$ copies/g tissue. **b** Score of viral RNA abundance (intensity within cells) detected using ISH-RNA (vRNA) targeting EBOV-VP40 or MARV-NP-VP35-VP40. **c** Virus-associated pathology score (VAPS), designated as co-localization of histopathological lesions with vRNA. Min-max normalization of IHC and VAPS scores are shown as percentages. Unavailable organs are depicted by grey. Lymph nodes (ovals) are mesenteric (histology) and cervical (q-RT-PCR). Abbreviations: Bld=Blood, Plac=Placenta, Spl=Spleen, Thy=Thymus and Bra=Brain. **d** TCID$_{50}$ virus titration (g tissue or ml of blood) of tissues sampled from pregnant bats ($n = 5$ biologically independent animals). Data shows mean ± SD from 4 technical replicates of each sample ($n = 8$ biologically independent tissue samples). **e** Photomicrographs of liver and placental tissues showing: ISH-RNA targeting EBOV-RNA of E03, E3o livers and E03 placenta; ISH-RNA targeting MARV-RNA of M05 placenta; and thin section electron microscopy (EM) of typical filamentous EBOV particles in the extracellular space of E3o liver and E03 placenta (right panels). Black arrows show enveloped virus particles of 85–125 nm in width, variable in length and that contain a rod-like nucleocapsid (see also Fig. S1). Scale bars: 500 μm (zoom-box: 20 μm) and EM: 200 nm. Source data are provided as a Source Data file.

suggested for BOMV in populations of AFBs[13,31]. Therefore, we hypothesized that EBOV would remain genetically stable in populations of $R_h$ and spillover based on its capacity to readily replicate in humans without requiring adaptation; as opposed to presenting high rate mutations in $R_h$ and spilling over only after a specific variant switches host species. To investigate and compare the extent of genomic adaptation required for filovirus host-shifts, we deep-sequenced and aligned the in vitro and in vivo passaged EBOV and MARV isolates used in this study (Fig. 5). We included the original human isolates passaged in, and potentially adapted to, Vero cells (EBOV_VC and MARV_VC, respectively) and in AFB-kidney cells (MoKi), as well as the re-isolated virus from in vivo infections used for subsequent cell passaging in Vero and MoKi cells. Throughout passaging, EBOV consensus sequences remained highly stable (Fig. 5b, Supplementary text), with only two single nucleotide polymorphisms (SNPs) detected compared to EBOV_VC. Both SNPs were within coding regions (VP24: A10831G and L: T14469C) and emerged only after in vivo infection, which, with one exception, persisted in subsequent passages. The A-to-G SNP within EBOV-VP24 generated a non-synonymous (dN) lysine-to-glutamic acid (K163E) residue substitution.

The VP24-K163E change is recurrent, arising either from EBOV human-to-human transmission[32] or through serial passaging in forced evolution models[33], although there is no described advantageous function. Mutations within EBOV-GP, however, are thought to be species-specific, driving EBOV tropism and human transmissibility[32]. Here, we detected no substitutions in EBOV-GP either at consensus or variant level (Fig. S9). In contrast, MARV isolates had 13 SNPs compared to MARV_VC: 10 within coding regions; 6 of which were dN (Fig. 5d–e, Fig. S10 and Table S7) and 4 of those within MARV-GP, suggesting virus adaptation to a novel host. Even after a comparatively stronger selection, MARV was unable to efficiently infect AFBs, whereas EBOV required little evolutionary change to infect not only human, NHP or AFB derived-cells, but also to efficiently replicate in the dam and infect its foetus.

## Discussion

Experimental inoculation of ERBs with MARV has helped to elucidate this filovirus' replication kinetics[18], the pathophysiologic consequences to infection in $R_h$[28], transmission pathways[5] and the $R_h$ immune response to infection[34]. However, productive EBOV infection in experimentally inoculated bats has been thus far unsuccessful[6,21]. The paucity of orthoebolavirus replication in ERBs suggests a strong species-specificity for a bat's role as a host of a specific filovirus. Herein, we show that infection of wild-caught AFBs with EBOV, but not RESTV, TAFV or MARV, display features expected of a reservoir-virus relationship. We report an in vivo experiment that has resulted in extensive replication of EBOV in 100% of inoculated bats, without inducing clinical pathology. Disease tolerance is well documented in many pathogen-reservoirs systems[28]. Likewise, MARV induces only mild subclinical pathology in its experimentally infected $R_h$. Here, we detected hallmark pathological findings of filovirus infection, reported in different host species[35]. Nonetheless, the subclinical pathology EBOV induces in AFBs is similar to what BOMV induces in AFBs[14] and

MARV in ERBs[28], in contrast to the clinical disease EBOV induces in humans and non-human primates[35]. The conspicuous high number of antigen-positive histiocytic cells in the spleen as well as Kupffer cells in the liver of EBOV-inoculated bats indicates the activation of the immune system in response to infection. The hypoalbuminemia detected in both EBOV and MARV cohorts could reflect an inflammatory process or hepatic damage, however, we would expect other hepatic markers to be altered or at least similar infection abundance and pathology between MARV and EBOV cohorts. Although these wild-caught bats showed no signs of disease during the quarantine period, it is possible that other infectious agents could be interfering with our results.

AFBs selectively supported EBOV replication and shedding compared to other filoviruses, resembling ERBs' unique permissiveness to MARV in contrast to orthoebolaviruses[21]. To assess the reservoir-competence of AFBs to filoviruses, we passaged the selected viruses in AFB-derived immortalized MoKi cells. Even though all viruses used in this study replicated efficiently in MoKi cell cultures and the same dose was used to inoculate all bats, there was a clear difference between the infection kinetics of EBOV and the other virus cohorts. Similar to MARV-$R_h$ infections[18,19], most of the EBOV-inoculated AFBs presented with a viremic phase, in stark contrast to the other virus cohorts. Similar to MARV tissue tropism in ERBs[19], EBOV-RNA was widely disseminated and still present at 10 dpi. This was clearly distinct in the pregnant females, which harboured the highest RNA and infectious virus titres in immune-privileged tissues, even at 10 dpi. RESTV replication was evidenced most consistently at the dermal inoculation site. In fact, RESTV-Ag was only detected in skin at 5 dpi, reflecting perhaps local replication or viral proteins lingering after the inoculation. TAFV showed no evidence of active replication in AFBs. Consistent with filovirus infection of ERBs[6,21], all bats in our study seroconverted by 10 dpi, despite the infection kinetic disparities between filoviruses. Still, lower Ab titres characterized the TAFV cohort. This nuanced disparate Ab-signatures corroborate that high active viral replication is conditional to elicit early immune responses against filoviruses in a reservoir-type host[6,21]. Due to the high susceptibility of AFBs to EBOV, it was not unexpected that RESTV infection was generally more abundant than the phylogenetically more distant MARV. It is unclear, however, why the more closely related TAFV[36] was not able to preferentially replicate in AFBs. The estimated slow evolution of filoviruses[37,38], evidenced by the long- and short-term conservation of EBOV sequences in humans[38,39], in persistent EBOV infection of AFB-derived cell cultures[40] and in geographically distant MARV isolates[41], was also demonstrated herein. The contrasting evolutionary signatures of the passaged viruses likely reflects the close relationship between AFBs and EBOV in comparison to MARV. Still, single-point mutations in RNA viruses can be decisive for the successful infection of a new host[42] and even slowly evolving viruses have a rich history of host-switching[43]; filoviruses have jumped between highly divergent fish, reptiles and mammal hosts in their ancient evolutionary history[43]. It seems possible therefore, that TAFV could have co-evolved with a genetically distant host. In fact, predictions of $R_h$ show that, contrary to the other orthoebolaviruses, TAFV (and also Bundibugyo virus)

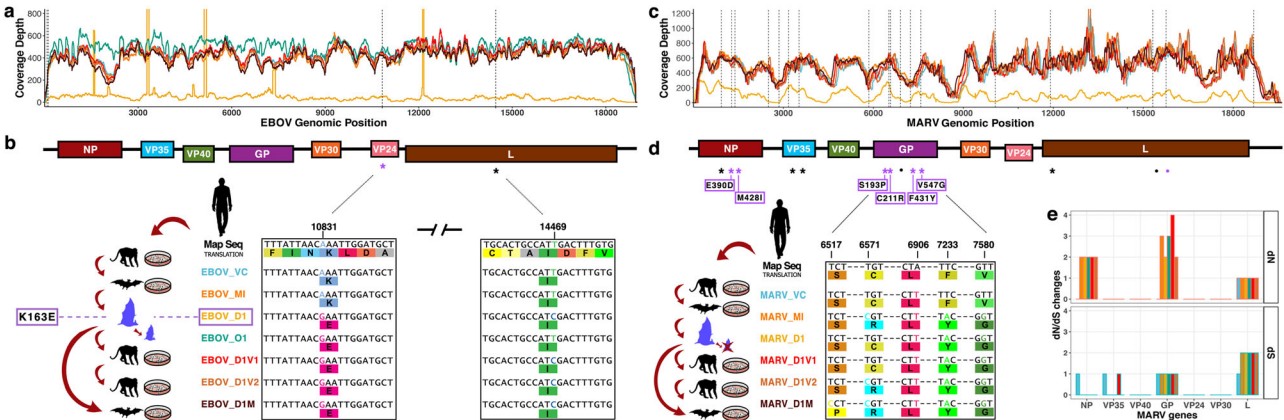

**Fig. 5 | Deep sequence assembly of filoviruses isolates used. a** Coverage of assembled EBOV isolates' sequences: isolate colour-coding shown in (**b**). Vertical dashed lines show sites of single nucleotide polymorphisms (SNPs) against the Map Seq. **b** EBOV genome schematic (top). SNPs against EBOV_VC are depicted as asterisks; both are within protein coding regions (coloured boxes). Assembled consensus sequences (below). Passaged isolates are depicted as cartoons; either in vitro (culture plate) or in vivo passage in bats (blue bats: dam and foetus). Sequence regions with SNPs and corresponding amino acid (aa) translations (coloured boxes below nucleotides) are highlighted within boxes. The purple box shows the non-synonymous aa substitutions (dN) arisen during in vivo passaging. **c** Coverage of assembled MARV isolates' sequences: isolate colour-coding shown in d. Vertical dashed lines show SNPs against MARV_VC. **d** Schematic of MARV genome

(top) and SNPs detected against MARV_VC (as above), shown as asterisks in black for synonymous changes (dS) or in purple for dN. Dots show SNPs that are unique against MARV Map Seq. Asterisks show SNPs positions in each gene and purple boxes enclose the residue changes. **e** dN and dS changes of the MARV passaged isolates against MARV_VC (colour-coding shown in **d**). Map Seq mapping sequence, VC original human isolate passaged in Vero cell, MI VC isolate passaged twice in AFBs kidney cells (MoKi) and used as inoculum for in vivo infection, D1 re-isolated virus from either kidney (EBOV) or placenta (MARV) of infected AFBs, O1 re-isolated EBOV from liver of infected foetus, D1V1 D1 isolate passaged in Vero cells once, and D1V2 = twice, D1M D1 isolate passaged in MoKi cells. Source data are provided as a Source Data file.

cluster with primate instead of bat hosts[37]. The virus-host specificity seen for MARV[5,9,44], Lloviu virus[45], BOMV[13,31] and here for EBOV, may reflect the host-shift of an orthoebolavirus' common ancestor from frugivorous to insectivorous bats (e.g. BOMV), some of which could have further adapted to a novel host taxon (e.g. TAFV).

Emphasizing the unique outcome of EBOV infection of AFBs, not only horizontal transmission was suggested by our findings, but vertical transmission was evidenced, which has not been reported previously for any filovirus in a bat host; not only was infectious EBOV isolated from most gastrointestinal and urinary tissues, but infectious virus shedding was also detected in secretions and excretions. Including contact with urine, high MARV loads detected in the oral mucosa of experimentally infected ERBs suggest that the virus can be transmitted bat-to-bat through biting. The orofecal route is the main model of MARV transmission among ERBs, with virus shedding peaking after the first week of infection[5,19,21]. Thus, it is plausible that similar transmission mechanisms of MARV and its $R_h$ are replicated by EBOV infection of AFBs. In fact, a super-shedding phenomenon reported for ERBs infected with MARV[5] is potentially reflected in the AFB-EBOV infection system. Yet, the seemingly heterogeneous individual susceptibility and shedding of the wild-caught AFBs were likely driven by the pregnant state of the bats. Indeed, changes in the immune environment during late pregnancy can drive viral reactivation and transmission[46]. However, only a fraction of infectious pathogens can traverse the placental barrier. Because the placenta is an immune-privileged site, vertical transmission has been associated with latency and chronic infections. Hence, it is a relevant mechanism for pathogen persistence in $R_h$ populations. Distinct from ERBs, AFBs congregate in small colony sizes (dozens to hundreds instead of thousands), forming maternity roosts[47]. Therefore, an increased susceptibility of virus reactivation and the synchronous parturition of highly infectious placenta and foetus could be an ancillary pathway for EBOV transmission and persistence in small colonies.

In summary, after more than 45 years of orthoebolavirus research, we demonstrate AFBs' usefulness to elucidate and mimic EBOV kinetics, transmission among conspecifics and possibly immunomodulation in a reservoir-type host. Furthermore, many of the features seen in

this experimental infection, in conjunction with previous epidemiological data and the harbouring of BOMV, indicate that AFBs may very well play a role in the sylvatic maintenance of EBOV. However, additional surveillance studies of, not only AFBs, but also other molossid populations are warranted to evidence EBOV natural infection in free-tailed bats. In fact, only after a large study in 2018, that surveyed 109 free-tailed bats including 52 AFBs, BOMV was discovered[13]; before that year AFBs only played a minor role in orthoebolavirus surveys. Co-circulation of RNA viruses, even within the same order as filoviruses, is not unprecedented in bats[48,49]. We highlight the importance that a human-derived EBOV is capable to infect AFBs and readily transmit through vertical and potentially horizontal pathways. We hypothesise that birthing seasons could not only drive EBOV maintenance in reservoirs, but also spillover to humans. However, blunt measures to stop virus transmission, such as culling, are not only detrimental to bat conservation, but also have been shown to be counterproductive[44,50]. Instead, targeted preventative measures, such as seasonal closure of cave exploration activities (e.g. corresponding to AFB birthing seasons) and raising public awareness of likely seasonal risks to spillover, could be implemented in the future.

## Methods

### Ethics and biosafety statement

Bat capture and work was performed following permission from the Laboratoire Central Vétérinair, Laboratoire National d'Appui au Développement Agricole (LANADA), Bingerville, Côte d'Ivoire (No. 05/virology/2016) and the Ministère des Eaux et Forêts (No. 0474/MINEF/DGFF/FRC-aska). The animal care and use protocol adhered with the Ethics Committee of LANADA and the National Ethics Committee for the Research (CNER). Consent to capture bats was obtained by the residence owners in Bregbo Village (colony location). Animal transportation was conducted following appropriate IATA Live Animals Regulations. Permission for housing of bats and infection experiments in the BSL-4 facility at the Robert Koch-Institute, Berlin, Germany, was granted by the Regional Office for Health and Social Affairs Berlin (LAGeSo, No. ZH 180 and No. G 0256/18, respectively).

## Sample size

Using a power analysis, a minimum number of 3 animals were planned for each experiment's time points. However, during the experiments for the first capturing period, a total of 10 animals were retrieved. Endpoint score for animal release during captivity and for euthanasia were pre-defined.

## Capture site and training

AFBs were trapped using 3 × 9 m mist nets positioned beside roofs of houses in Bregbo village (geographical coordinates: N 05° 18.229'; W 003° 49.396'), Côte d'Ivoire. Two trapping trips were required for the experiments: May and October 2019. The bats were immediately transported in cotton bags to a flight cage assembled at LANADA (Fig. S11) for a minimum 4-week quarantine. Here, bats were inspected and only adults (neither pregnant nor lactating females) were selected for the study. Species was determined by visualization of morphological features and subsequently confirmed by sequencing of a 241 bp fragment of the *cytochrome b* gene (Supplementary Data 1)[51]. Briefly, a DNeasy Blood & Tissue Kit (Qiagen) was used to extract DNA from the saliva of captured bats and a Platinum Taq DNA Polymerase (Invitrogen) was used for *cytochrome b* amplification. For each PCR reaction, 5 μl of the DNA sample was added to 20 μl master mix containing: 0.75 μM forward/reverse primers (forward: ccccHccHcaYatYaaRccMgaRtgata; reverse: tcRacDggNtgYcctccDattcatgtta), 2.5 μl 10X buffer, 0.2 μl Platinum Taq DNA Polymerase, 2 mM Mg, 2.5 mM dNTPs and 12.8 μl nuclease-free water. Samples were incubated for 10 min at 95 °C followed by 45 cycles of 30 sec at 95 °C, 30 sec at 55 °C and 30 sec at 72 °C, followed by an extension of 5 min at 72 °C.

## Experiment design—Stage 1: Diet adaptation and quarantine

Age, sex, phenological parameters and weight were recorded from the day of capture. Sex was determined by direct visualization of external genitalia and age (described as neonate, juvenile, subadult and adult) was assessed according to body weight, genital and teat development as well as qualitatively by the degree of mineralization of the epiphyseal-diaphyseal fusion area[52]. Bats were co-housed independent of sex. Weight was recorded nightly before feeding using a spring scale (Pesola®). The health of bats that sustained >20% body mass loss was evaluated for continuation of feeding training or release. A diet of mealworms (*Tenebrio molitor* Linnaeus, 1758) was provided to the animals. Water was not supplemented during captivity (quarantine or experimental housing). According to the feeding progress, bats were evaluated from a scale of 0 to 6 (Table S1–S2). Bats were hand-fed mealworms using disposable plastic forceps from the beginning of the quarantine. Until transportation, hanging dishes were filled with worms for *ad libitum* feeding. Animals that self-fed from hanging dishes were selected for transport for infection studies. A box (55 × 45 × 52 cm) complying to IATA Container Requirement 77 for bats was used to transport a maximum of 25 animals. For the May 2019 capture, the initial date of transportation was postponed 10 weeks due to airline policy changes. This resulted in a route detour and an additional travel day (from an estimated planned travel time of 38 h, to approximately 60 h).

## Experiment design—Stage 2: acclimation and inoculation

All infectious work was performed in the BSL4 laboratory at the Robert Koch-Institute. The experiments were performed during two periods: September and November 2019. Between the two capture efforts, a total of 33 animals were shipped and 32 arrived to our facility healthy; one animal was unfit for the experiment, likely as a result of to the extended captivity and air transportation: 7 males and 4 females from the capture in May (1 male euthanized after arrival); 19 males and 3 females for the capture in October. The bats were grouped in stainless steel mesh flight cages (1 × 2 × 2 m) attached to a darkened sleeping box (0.4 × 0.3 × 0.4 m) in a climate-controlled room (27 °C ± 1 °C;

humidity of 70% ± 5%; 12 h light-dark cycle) for an acclimation period of at least 7 days before inoculation (Fig. 1a and Fig. S3). Mealworms were offered in several hanging dishes, *ad libitum*, in the flight cage. Bats were divided in groups according to the virus inoculum, each of which were later subdivided into two time point cohorts according to the euthanasia day post-virus inoculation (dpi): 5 and 10 dpi. Virus groups had disproportionate sex composition. To allocate at least one female per virus group, bats were randomly assigned to groups, separately by sex (Fig. S3). The animals selected for a virus group were co-housed until experimental endpoint.

## Virus inoculation

EBOV-Makona-C05 (GenBank MG572234.1), MARV-Musoke (GenBank: DQ217792.1), RESTV-USA (GenBank: KY798006.1) and TAFV (GenBank: FJ217162.1), initially isolated on Vero cells, were passaged twice in a previously established AFBs kidney (MoKi) cell line[51] for virus adaptation. The four viruses used in the experimental inoculations adapted successfully in MoKi cell cultures and all replicated to similar titres (Table S3). After tissue culture infective dose 50% ($TCID_{50}$) titrations the MoKi-passaged stocks were then diluted in sterile Dulbecco's modified Eagle's medium (DMEM, GE Life Science) containing 5% of foetal bovine serum, 1% penicillin/streptomycin and 1% L-glutamine (FBS 5%, P/S 1%, L 1%), to a concentration of $1 \times 10^6$ $TCID_{50}$/ml (equivalent to $1 \times 10^8$ viral RNA copies/ml).

Under isoflurane anaesthesia, each bat was inoculated via subcutaneous (s.c.), intraperitoneal (i.p.), oral and nasal routes, with a total inoculum dose of $6 \times 10^4$ $TCID_{50}$ in a volume of 60 μl (10 μl oro-nasal routes, and 20 μl s.c. and i.p.) to avoid potential route-dependent barriers. Ancillary analyses of virus replication include an effective-inoculum-dose estimate (Fig. S4). To assess true virus replication in tissues, the undiluted inoculum dose was deemed artificial. To address this, although still inaccurate, the total inoculum dose (RNA copies in the inoculum volume), was divided by the average total animal blood volume (80 ml/kg of body weight).

Bat health and food intake was scored daily to assess the development of clinical signs, maintain hygiene of the cage and collecting faecal and urine samples.

## Specimen collection

Blood (up to 20 μl) and oral swabs were obtained during captivity in Côte d'Ivoire, to assess potential previous exposure to filoviruses. Oral swab samples were used to test exposure and active infection of lyssaviruses and filoviruses. Briefly, 7 genotypes within the *Lyssavirus* genus were screened using a nested RT-PCR[53]; filovirus active infection was screened using a RT-qPCR assay that has been validated to detect EBOV, SUDV, RESTV, TAFV, BDBV and MARV[54]. Previous exposure to filoviruses was screened using a previously validated Luminex assay[26]. Blood was collected from the cephalic wing vein using a sterile needle and a micropipette (p-100) and placed in 300 μl gel tubes (BD Microtainer®) for serum separation (5 min centrifugation at 8000 x g). Pre-inoculation, at 0 dpi, and post-inoculation, at necropsy, oral and rectal samples were taken under isoflurane anaesthesia using polyester-tipped swabs (Copan FLOQSwabs 80501CS, Mast-Group) stored in 500 μl of phosphate-buffered saline (PBS, GE Life Science). Post-inoculation blood was obtained once per animal at euthanasia via cardiac puncture and exsanguination. Group faeces and urine were collected daily. For this, faecal pellets were collected in empty 2 ml tubes and urine was collected using a dry polyester-tipped swab, stored in 500 μl of PBS. The collected faecal pellets were weighted, diluted 1:10 in PBS and centrifuged for 10 min at $12,000 \times g$ for supernatant collection. Necropsies were performed immediately after exsanguination and cervical dislocation. Samples of 20 organ tissues, plus blood, oral and rectal mucosal samples (Fig. 2a, Table S4) were collected in triplicates, when sufficiently available, for downstream analyses. For virus isolation, approximately 100 mg of each organ

sample was stored at -80 °C in empty vials. For viral RNA quantification, approximately 30 mg of tissue was placed in 600 µl of RLT buffer (Qiagen), or 140 µl of liquid samples were added to 560 µl of AVL buffer (19073, Qiagen) and frozen at −80 °C until further processing. For histology, pre-selected areas of organs that included most tissue types or regions within, were fixed in 10% formaldehyde (ROTI®Histofix 10 %).

### Real-time quantitative reverse-transcription PCR (RT-qPCR)

Tissue samples were homogenized in RLT buffer using a stainless-steel bead and a Tissuelyser II (Qiagen) homogenizer, for 10 min at 30 Hz. Samples were then centrifuged for 10 min at $6000 \times g$ and the supernatant was inactivated in 600 µl of 70 % ethanol per 30 mg of tissue. Liquid samples in AVL buffer were inactivated in 560 µl of 100% ethanol. Viral RNA was extracted (50 µl elute) using a QIAmp Viral RNA Mini Kit (Qiagen) or a RNeasy Mini Kit (Qiagen) following the manufacturer's instructions. AgPath-ID One-Step RT-PCR mix (4387391, Thermo Fisher) was added to 3 µl of extracted RNA sample to be quantified using an Applied Biosystems 7500 thermocycler and an ABI 7500 Real-Time PCR systems software v2.3. Primers and probes targeting either the VP30 or L-gene were added to 22 µl of master mix for total volume of 25 µl and run with the specified cycling conditions (Table S5–S6). All samples were measured in duplicate. To quantify the RT-qPCR cycle threshold (Ct) values, we used standard curves generated using virus-specific in vitro transcripts of known concentrations (10 to $10^7$ copies). Values with Ct values ≤ 40 were deemed positive.

### Virus isolation

Virus isolation and $TCID_{50}$ titration was performed using frozen samples from selected tissues that had virus RNA copies/g above a cut-off of $>5 \times 10^5$ or Ct values < 30, or were of foetal origin and available for titration. Tissue samples were homogenized in 500 µl of medium (DMEM FBS 5%, P/S 1%, L 1%) for 10 min at 30 Hz. Following standard titration protocols[55], sample supernatants (at an initial dilution of 1:10) were 10-fold serially diluted and assayed in quadruplicate. Dilutions were incubated (37 °C with 5% $CO_2$) for one hour in 96-well microtiter plates (Nunc, Thermo Scientific) seeded with $3 \times 10^4$ MoKi cells/well. Working medium was used for negative controls. Cytopathogenic effect (CPE) was monitored after 14 days. Virus titre was calculated by assessing the number of CPE-positive wells per dilution and applying the Spearman-Kärber method. For samples in which CPE was not evidently induced by virus replication (e.g. stomach, gallbladder), virus replication was confirmed by indirect immunofluorescence assays using in-house monoclonal antibodies against EBOV or MARV Nucleoprotein (NP) and by RT-qPCR quantification.

Selected samples that were below our cut-off: that had virus RNA copies/ml $<5 \times 10^5$ (or Ct values < 30) but $>2 \times 10^2$ copies/ml, such as faeces, bladder and mucosal swab samples (oral and rectal), were homogenized as previously described. Samples were then incubated in 25 cm² flasks for further passaging and expansion in 75 cm² flasks. Virus replication was assessed using RT-qPCR quantification.

### Serology

A previously established in-house Luminex multiplexing assay[26] was used to detect antibodies to filoviruses in serum samples of bats before and after inoculation. The previously validated positive control serum samples, derived from AFBs inoculated with EBOV ($n = 1$) or MARV ($n = 1$) virion-like particles (IBT Bioservices #0550−001, 1.54 mg/ml and #0566−001, 3.725 mg/ml, respectively), were used as positive controls[26]. Due to the different sample inactivation process used herein (infectious samples inside the BSL-4 compared to non-infectious control samples), we treated all samples and controls following the same protocol and used a standard cut-off method. Briefly, $1 \times 10^6$ activated magnetic beads were coupled with EBOV or MARV-NPs. Beads uncoupled to any protein and beads coupled to bacterial

lysate were included in each sample to control for, and to subtract unspecific binding (background correction). Previous to the serological analyses, infectious serum samples and controls were diluted 1:25 in inactivation buffer (0.5% Triton X-100 and 0.5% Tween-20) and incubated for 30 min at 60 °C. To avoid unspecific binding after inactivation, samples were re-diluted in low binding buffer (Low Cross Buffer, Candor) with 5% E. coli lysate to achieve total dilution of 1:200. Specific fluorescence wavelengths, of each antigen-coupled bead, were measured with the Luminex L200 analyser (DiaSorin) and the Bio-Plex Manager Software 6.1. Results show the mean fluorescent intensities (MFI), after background correction of all samples run in duplicates, extrapolated from 100 beads counted per sample/well. Positive and negative controls were included in each plate. Serum of mock-inoculated bats ($n = 2$) as well as the control bats kept in BSL-4 conditions ($n = 8$) and blood taken pre-infection ($n = 22$) were used to estimate the assay cut-off: average MFI of non-infected bats plus 3 times the standard deviation (SD). The assay cut-off resulted in an MFI of 422. Previously validated positive controls for EBOV and MARV resulted in MFIs of 8938 and 11,126, respectively.

### Histopathology, Immunohistochemistry and RNAscope® ISH (ISH-RNA)

Samples of organs (heart, spleen, lymph nodes, tonsils, larynx, trachea, lung, liver, pancreas, salivary glands, oesophagus, tongue, stomach, small intestine, large intestine, kidneys, skeletal muscle, brain, testes, uterus, adrenal glands, thyroid, skin) were fixed in 10% formaldehyde (ROTI®Histofix 10%) for 7 days and were later trimmed, processed and embedded in paraffin. Serial tissue sections (3 µm) were either stained with haematoxylin and eosin (HE) or were submitted for immunohistochemistry. All tissues were examined microscopically by study-blinded veterinary pathologists.

EBOV and RESTV immunoreactivity was detected using a rabbit anti-EBOV-VP40 antibody (provided by Yoshihiro Kawaoka, University of Wisconsin-Madison, USA) at a 1:2000 dilution. MARV immunoreactivity was detected using a rabbit anti-MARV-NP antibody (provided by Ayato Takada, Hokkaido University, Japan) at a 1:1000 dilution. The secondary detection antibody was a horse anti−rabbit IgG/polymer "ready-to-use" system from Vector Laboratories ImPress VR (catalogue #MP-6401-15, lot number: ZH1216). Tissues were then processed for IHC using the Discovery Ultra automated processor (Ventana Medical Systems) with a Roche Tissue Diagnostics Discovery Purple Kit (catalogue #760-229).

For ISH-RNA, probes targeting EBOV-Makona-VP40 (Cat No. 450581) and MARV-NP-VP35-VP40 (Cat No. 527301) genes were purchased from a commercial source (Advanced Cell Diagnostics [ACD], Newark, CA). To assess assay specificity, a negative control probe specific for Bacillus subtilis Dapb mRNA was used. To test sample RNA integrity, a positive control was designed by sending sequences of Mops condylurus' (Ppib) gene to ACD for design and manufacture. The ISH-RNA assay was performed using an RNAscope 2.5 HD-Brown Detection Kit (ACD). Briefly, deparaffinized sections were subjected to target retrieval for 15 min at 98-102 °C in 1X Target Retrieval Solution, dehydration in 100% ethanol for 10 min, and Protease Plus treatment for 30 min at 40 °C in a HybEZ™ oven (ACD). Slides were subsequently incubated with a ready-to-use probe mixture for 2 h at 40 °C in the HybEZ™ oven, and the signal was amplified using the recommended set of amplifiers (AMP1-6). The signal was detected using a BROWN solution (DAB-A: DAB-B in a 1:1 ratio) for 1–10 min at room temperature. Slides were counterstained with 50% haematoxylin staining solution for 2 min. A wash step with 0.02% ammonium hydroxide in water was done to achieve a blue colouring.

### Antigen-associated pathology scoring

IHC stained tissues that were positive for filovirus antigen were further examined in parallel with HE-stained slides to evaluate lesions that

co-localized with the viral antigens. The antigen staining intensity within reactive cells was scored from 0 to 3. For direct comparison of the degree of associated pathological lesions induced by the virus, a virus-associated pathology score (VAPS) was evaluated in a 0–5 scale, where 0 = no changes, 1 = mild, 2 = mild to moderate, 3 = moderate, 4 = moderate to severe and 5 = severe changes. Due to the mostly patchy overall distribution of IHC-positive cells (single cells versus localized accumulation versus generalized distribution) we refrained from scoring positive IHC results for entire organ sections.

## Thin section electron microscopy

Formalin-fixed tissue samples were post-fixed in a mixture of formaldehyde (1%) and glutaraldehyde (2.5 %) in Hepes buffer (0.05 M, pH 7.2) for several days. After washing out the fixative with Hepes buffer, post-fixation in osmium tetroxide, followed by bloc-contrasting in tannic acid and uranyl acetate, dehydration and embedding in Epon was conducted according to a standard protocol[56]. Ultrathin sections were produced at a thickness between 60-70 nm using an ultra-microtome (UC7, Leica Microsystems) and contrasted with uranyl acetate and lead citrate. Transmission electron microscopy (Tecnai Spirit BioTwin, Thermo Fisher) was used at 120 kV acceleration voltage to inspect the sections. Images were recorded with a CMOS camera (Phurona, EMSIS, Radius software version 2.1) mounted to the side-entry port. Magnification calibration was performed by using the MAG*I*CAL (EMS) reference standard.

## Blood chemistry

Sera were tested in a Fuji DRI-Chem NX500i platform (FUJIFILM Europe GmbH). Serum samples were diluted 1:5 with isotonic saline solution. Diluted samples were used to measure gamma glutamyl transpeptidase (GGT), alanine transaminase (ALT/GPT), alkaline phosphatase (ALP) and creatine phosphokinase (CPK) parameters. Undiluted sera were used to measure total bilirubin (TBIL), total cholesterol (TCHO), serum glucose (GLU) and albumin (ALB). Different dilutions were tested with sera obtained from captive AFBs (in Côte d'Ivoire). Due to the heterogeneity of the data points, we only compared filovirus-inoculated bat cohorts with the combined control groups of wild caught-bats and control bats (mock-inoculated or captive bats in the BSL-4). Serum of 17 wild caught-bats and 10 control bats (mock-inoculated and captive bats in the BSL4) were measured. Parameters that were out of range were re-tested using increased or decreased serum dilutions. Not all parameters were available for all bats. The parameter with the lowest sample size was ALB and the EBOV cohort ($n = 3$).

## High-throughput sequencing of EBOV and MARV isolates

To evaluate possible genomic adaptation resulting from the experimental host-shifts, we aligned the filoviruses' genome sequences that infected different host species to complete the passaging history, both in vitro and in vivo (Fig. 4 and Fig. S9–10). After target enrichment using in-solution hybridization capture, we deep-sequenced the viruses grown in our laboratory on Illumina® platforms, and mapped resulting reads against the sequences of the original human-derived isolates (GenBank sequence accession number MG572232.1 and DQ217792.1) to assemble whole genomes.

For the library preparation, we built dual-indexed libraries for Illumina® sequencing of the EBOV and MARV isolates (sequences to be released): The isolates included were: 1) the human original isolate passaged in Vero cells: EBOV_VC (Accession Number: SAMN36810525) and MARV_VC (Accession Number: SAMN36810532), which was then 2) passaged twice in MoKi cells: EBOV_MI (Accession Number: SAMN36810526) and MARV_MI (Accession Number: SAMN36810533): this was used as inoculum in the in vivo infection experiment. 3) The re-isolated virus obtained from kidney tissue of E03: EBOV_D1 (Accession Number: SAMN36810527) or the placenta of DM1: MARV_D1 (Accession

Number: SAMN36810534) and in case of EBOV 4) the virus extracted from the liver tissue of E03's foetus: EBOV_O1 (Accession Number: SAMN36810528). We then used EBOV_D1 or MARV_D1 to infect Vero cells for 5) one: EBOV_D1V1 (Accession Number: SAMN36810529) and MARV_D1V1 (Accession Number: SAMN36810535) and 6) two: EBOV_D1V2 (Accession Number: SAMN36810530) and MARV_D1V2 (Accession Number: SAMN36810536) passages. 7) We also used EBOV_D1 or MARV_D1 to infected MoKi cells: EBOV_D1M (Accession Number: SAMN36810531) and MARV_D1M (Accession Number: SAMN36810537) for one passage.

To convert the nucleic acid extracts of the cell-culture supernatants or tissues to dual-indexed libraries, we first performed DNase treatment on 20 µl for MARV extracts, 25 µl for EBOV extracts, and in case of EBOV_O1, 40 µl supernatant using the TURBO DNA-free™ Kit (Ambion) and cleaned up the reactions with the RNA Clean & Concentrator Kit (Zymo Research). The purified RNA served as template to generate cDNA using the SuperScript™ IV First-Strand Synthesis System (Invitrogen), and was subsequently turned into double-stranded DNA with the NEBNEXT® mRNA Second Strand Synthesis Module (New England Biolabs). The DNA was then purified using MagSi-NGSprep Plus Beads (Steinbrenner Laborsysteme) and eluted in 20 µl Tris-HCl (10 mM) EDTA (1 mM) Tween20 (0.05%) buffer (TET). For all following steps ca. 200 ng of chicken DNA were included as a control sample. Prior to library preparation, the sample volume was filled up to a volume of 130 µl TET and fragmented using a Covaris S220 Focused-ultrasonicator with settings to generate 400-bp fragments (intensity = 4, duty cycle = 10%, cycles per burst = 200, treatment time = 55 s, temperature = 7 °C). The fragmented DNA was concentrated with the MinElute PCR Purification Kit (Qiagen) and eluted in 51 µl TET. 1 µl was used to measure the DNA concentration on a Qubit™ 3 Fluorometer (Invitrogen) using the Qubit™ dsDNA HS Assaykit (Invitrogen). The DNA concentration in EBOV_MI, MARV_VC, MARV_MI, MARV_D1V1, MARV_D1V2 and MARV_D1M was too low to measure, whereas MARV_D1 contained 10.8 ng DNA, the other EBOV samples contained between 2.05 and 5.43 ng DNA, with the exception of EBOV_O1, which contained a total of 452 ng. The remaining 50 µl served as input for library preparation with the NEBNext® Ultra™ II DNA Library Prep Kit for Illumina® (New England Biolabs). Based on the amount of input DNA, the adapters were used undiluted for EBOV_O1, MARV_D1, and chicken DNA, diluted 1:10 for EBOV_D1V, EBOV_D1V2, and 1:25 for EBOV_VC, EBOV_MI, EBOV_D1, EBOV_D1M, and all other MARV samples. Size-selection was performed for EBOV_O1 and chicken DNA, for all other samples the input was too low for size-selection. All clean-up steps during the library preparation were conducted using MagSi-NGSprep Plus Beads (Steinbrenner Laborsysteme). Dual-indexes were added to the libraries during 6 PCR-cycles for EBOV_O1, MARV_D1 and chicken, 11 cycles for EBOV_D1V, 12 cycles for EBOV_VC, EBOV_D1, EBOV_D1M and EBOV_D1V2, 13 cycles for MARV_VC, MARV_MI, MARV_D1V1, MARV_D1V2 and MARV_D1M and 14 cycles for EBOV_MI using the NEBNext® Multiplex Oligos for Illumina® (New England Biolabs). The dual-indexed libraries were quantified using the KAPA Library Quantification Illumina Universal Kit (Roche). The EBOV_D1M library had a low concentration and was therefore amplified for an additional 4 cycles with the KAPA HiFi HotStart ReadyMix (Roche) and Illumina adapter-specific primers to reach a concentration sufficient as input for in-solution hybridization capture.

For the in-solution hybridization capture, EBOV and MARV DNA were enriched following the myBaits Hybridization Capture for Targeted NGS protocol (Version 4.01) using custom-made RNA baits (120 nucleotides long, 2-fold tiling; Arbor Biosciences) that cover representative genomes of *Orthoebolavirus zairense* (KC242801), *Orthoebolavirus sudanense* (KC242783), *Orthoebolavirus restonense* (NC_004161), *Orthoebolavirus taiense* (NC_014372), *Orthoebolavirus bundibugyoense* (KC545395) and *Orthomarburgvirus marburgense*

(FJ750956). Only a fourth of the recommended bait input volume was used. For MARV, we pooled indexed samples for capture depending on their Ct-values in the MARV-specific qPCR (pool 1: MARV_D1, MARV_MI, MARV_D1M, pool 2: MARV_VC, MARV_D1V, MARV_D1V2). For EBOV, we prepared separate capture reactions for each sample using ca. 150 ng library as input. We performed two 24h-long rounds of hybridization capture at a hybridization temperature of 65 °C. After both rounds of capture, the capture products were amplified using the KAPA HiFi HotStart ReadyMix (Roche) and Illumina adapter-specific primers (first round 12-25 PCR cycles, second round 12-17 PCR cycles), quantified using the KAPA Library Quantification Illumina Universal Kit (Roche), and cleaned up using the MinElute PCR Purification Kit (Qiagen) with an elution volume of 10 μl. We used 7 μl of the first-round product as input for the second round of capture. The second-round product was quantified and diluted to 4 nM for sequencing on an Illumina® MiSeq platforms, and to 1 nM for Illumina® NextSeq and iSeq.

For the sequencing, all EBOV capture products except for EBOV_O1 were first sequenced on an Illumina MiSeq platform (V3 chemistry, 2 × 300-bp reads), then re-sequenced on an Illumina® NextSeq platform using v2 chemistry (2×150-cycle) to increase depth. EBOV_O1 and all MARV samples were sequenced on an Illumina® iSeq platform using iSeq 100 i1 Reagents (2 × 150-cycle).

For the genome assembly, sequencing reads were filtered (adapter removal and quality filtering) with Trimmomatic[57] (settings: LEADING:30 TRAILING:30 SLIDINGWINDOW:4:30 MINLEN:40). Filtered read pairs were merged using ClipAndMerge[58], and merged, unmerged and unpaired reads for each sample were combined into a single file, which was mapped to a *Orthoebolavirus zairense* Makona strain (MG572232) or to *Orthomarburgvirus marburgense* Musoke strain (DQ217792.1) using BWA-MEM[59]. The mapping files were sorted and duplicates were removed with the tools SortSam and MarkDuplicates from the Picard suite[60]. We then used Geneious Prime[61] (v2021.2.2) to assemble consensus genomes, calling bases with a minimum coverage of 20x and a 50% majority. For EBOV_D1 and MARV_D1, we had fewer reads on target and therefore called the bases at 3x and a 50% majority. To compare the genomes, we aligned the consensus sequences using MAFFT v7 implemented in Geneious[62] and visually inspected the alignment. For EBOV samples, we also performed minority variant (min-v) analyses in Geneious Prime. For the samples with good coverage, we called variants with a minimum coverage of 100 and a minimum variant frequency of 0.1. For EBOV_D1, we called variants with a minimum coverage of 50 and a minimum variant frequency of 0.1 though it has to be noted that the sequencing depth of coverage of this genome may not be sufficient for meaningful results. In Fig. S9–10, we show variant percentages for bases with at least 50 sequences of depth. Assemblies had a coverage depth over 400 on average, except for EBOV_D1 that had a mean depth of ~70 but spiked over 5000 sequences in some regions (Fig. 5a).

## Statistical analysis

Microsoft Excel 16.0 was used for data output collection. All statistical analyses were done using R[63] (v4.1.2) and the packages dplyr, rstatix, Rmisc and dunn.test. Due to sample size, statistical tests and descriptive statistics, such as mean (M) and standard deviations (± SD), or median (Mdn) and interquartile range (IQR) values, were only performed in groups with at least 3 animals (n = subsample size). No outliers were excluded from the analyses. Statistically significant differences between groups ($\alpha = 0.05$) were assessed with two-tailed tests. After determination of distribution normality using qqplot visualization and a Shapiro-Wilk test (with and without data transformation), either the non-parametric Kruskal-Wallis H-statistic or a one-way ANOVA F-statistic (F) was used to compare sample distributions of two or more independent sample groups (e.g. daily body weight change between captivity periods, or blood chemistry parameters between cohorts). Kruskal-Wallis chi-squared ($\chi^2$), degrees of freedom

(df) and effect size (eta2[H] or $\eta^2$) are reported. Pairwise comparisons, Dunn test (one-sided) and Tukey's HSD were used as post-hoc analysis using Bonferroni for multiple comparison adjustment and reported as *p*-adjusted. The Friedman test was used to compare virus groups (RNA/g titres) of related data (tissues were used as blocks and only paired organs were included); Friedman's chi-squared (Q), Kendall's effect size (W) and df are reported. Spearman's rank correlation Rho test was selected to detect data associations.

## Visualization

Visualization of statistical analyses were done using the package ggplot2 within R[63]. Histopathology pictures were processed using ImageJ version 1.52a. For other visualizations, Adobe Photoshop and Adobe Illustrator CS6 (64 bit) were used.

## Reporting summary

Further information on research design is available in the Nature Portfolio Reporting Summary linked to this article.

## Data availability

Source data are provided with this paper. The main data supporting the results in this study are available within the main manuscript and the supplementary materials. The accession codes for the previously published GenBank sequences used for the genomic analyses herein, are: MG572232.1, DQ217792.1. The myBaits Hybridization Capture for Targeted NGS protocol covers the following previously published GenBank sequences: KC242801, KC242783, NC_004161, NC_014372, KC545395 and FJ750956. The resulting raw aligned bam files generated in this study are available from the Genbank BioProject accession number PRJNA1001398 and Sequence Read Archive (SRA) accession number SRP453002. The BioSamples and SRA individual accession codes are: SAMN36810537 and SRX21227671, SAMN36810536 and SRX21227670, SAMN36810535 and SRX21227669, SAMN36810534 and SRX21227679, SAMN36810533 and SRX21227678, SAMN36810532 and SRX21227677, SAMN36810531 and SRX21227676, SAMN36810530 and SRX21227675, SAMN36810529 and SRX21227674, SAMN36810528 and SRX21227673, SAMN36810527 and SRX21227672, SAMN36810526 and SRX21227668, SAMN36810525 and SRX21227667. The individual accession codes are also included in the main manuscript. Source data are provided with this paper.

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

## Acknowledgements

We are grateful to Romeo Yao N'Doua for vital technical support. We thank Doris Krumnow, Monique Schmückert and Dagmar Viertel for excellent technical assistance. Most importantly, we thank World Courier Berlin office for their continuous support and persistence to solve the ever-increasing problems of animal transportation for research purposes. This work was supported by the Robert Koch-Institute.

## Author contributions

Conceptualization: S.A.R.-S., A.K. Methodology: S.A.R.-S., G.W., A.D., V.K., M.B., M.L., E.C.H., A.K. Investigation: S.A.R.-S., G.W., A.D., V.K., M.B., K.H.-K., N. Ki., M.L., N. Kr., A.L., U.V., D.M.W., A.W., D.P.S., J.P., L.S., E.C.H., A.K. Visualization: S.A.R.-S., G.W., M.L. Supervision: A.K., L.S. Writing—original draft: S.A.R-S. Writing—review & editing: S.A.R-S., G.W., A.D., V.K., M.B., K.H.-K., N. Ki., M.L., N. Kr., A.L., U.V., D.M.W., D.P.S., J.P., L.S., E.C.H., A.K.

## Funding

## Competing interests

The authors declare no competing interests.

## Additional information

**Peer review information** : *Nature Communications* thanks the anonymous reviewer(s) for their contribution to the peer review of this work. A peer review file is available.

