## [Peer Review File · Nature Communications]

Selective replication and vertical transmission of Ebola virus in experimentally infected Angolan free-tailed batsREVIEWER COMMENTS

Reviewer #1 (Remarks to the Author):

Thank you for the opportunity to review this interesting and relevant paper. The authors describe experimental infection of an insectivorous bat species, the Angolan free-tailed bats and provide evidence that this species may be a potential reservoir of EBOV (previously Ebola Zaire). In addition, does the paper also provide important information on the husbandry of insectivorous bats in experimental infections that can inform future studies. The paper is well written, and I only have a few minor comments below;

General comments;

1. The authors should update the filovirus taxonomy, especially genus and species names and use of italics according to the recent publication; Biedenkopf et al., 2023, Archives of Virology (2023) 168:220, <https://doi.org/10.1007/s00705-023-05834-2>
2. Ebolavirus has never been detected in the Angolan free-tailed bat, and this species has only been implicated through circumstantial evidence. Bombali virus on the other hand, has been detected. It would have added a lot of value to include Bombali virus in this study for comparative purposes. If not now, maybe in future work.
3. Also linked to the point above. If the Angolan free-tailed bat is a potential reservoir for EBOV, why was it not detected in surveillance studies. Several surveillance efforts reported the presence of Bombali virus. I would have expected EBOV positives also if a reservoir. Can the authors please address this point in the discussion?

Specific comments;

Line 36-39 and Table 1; Virus isolation was not attempted for all PCR positive samples. Can the authors indicate why e.g. 10dpi for EBOV?

Line 292-294; Is it possible to indicate that BOMV also have a slow evolutionary clock based on very limited detections of this virus?

Line 300; It indicates "original human isolates passages in Vero cells". Is this not already adapted virus and how many times was it passaged?

Line 327; I suggest changing: "not the filoviruses" to the specific ones included in this study. Not all filoviruses were included.

Line 407-412; I find these last sentences very confusing. Why link the results to culling? Also, expand on the link with birthing seasons and specifically what targeted preventative measures can be used.

Line 434; Add the country, Ivory Coast

Line 441; Why only sequence such a short fragment? Also, add the GenBank accession number of these sequences for reference and indicate what analyses were used to confirm species identification. There have been several reclassifications in the Mops genus recently, and it needs to be clearly demonstrated that the species used is indeed *Mops condylurus*. I suggest adding this to the supplementary information to prove this point.

Line 464, Stage 2; Please comment on how many animals were transported and how many survived.

Line 505; Add more specific results on the testing of filovirus previous exposure. Was a full panel of filovirus antigens used for this purpose? Any testing for other virus infections?

Table 1; page 30

Reviewer #2 (Remarks to the Author):

The manuscript by Riesle-Sbarbaro et al. describes experimental filovirus infection of Angolan free-tailed bats. The manuscript is well-written and provides a highly-detailed experimental infection model with a broad analysis of the infectious process. The work strongly supports the hypothesis that this species of bat is a likely reservoir host of Ebola virus, a question that has remained enigmatic for a couple of decades. This manuscript is a significant step in understanding the ecology of Ebola virus.

The description for adapting the bats to feeding upon mealworms is excellent and may provide a guide for use of other insectivorous bats of importance to virus research.

This reviewer only has a few suggestions for the authors to consider. First, were any of the polymorphisms identified in figure 5 present in the stock virus preparation? Deep sequencing is unlikely to detect such rare variants, which could have been selected upon passage in cell culture or bats. Were multiple (e.g., 3) passages independently conducted to

determine if the polymorphisms were identical? If the same polymorphisms persisted, it suggests they were present in a low number of viruses in the original stock. PCR with a primer spanning the polymorphism(s) could also address this question.

Does the qPCR assay discriminate between viral genomic and mRNA?

With the suggestion that macrophages may be susceptible (SF6j) as any attempt made to generate macrophages from bat PBMC or bone marrow to determine if they were susceptible to infection?

The authors should note in the discussion that a potential issue with using wild-caught bats is the possibility they may unknowingly harbor other infectious agents or be impacted by environmental factors that could influence the results and interpretations of the study.

A few minor considerations:

Line 31. Probably should change to “Bombali virus” without italics.

L297. Change to “specific variant switches host species.” (I believe this is the authors’ intent)

L392. Change to “infectious pathogens can traverse”

L939. Change “Since” (passage of time) to “Because”.

L401. Change “utility” to “use”.

Reviewer #3 (Remarks to the Author):

Although Ebola virus (EBOV; family Filoviridae, genus Orthoebolavirus) was discovered almost 50 years ago, its natural reservoir has not been identified. Cumulative PCR, serological, and circumstantial epidemiological evidence indicates that bats are natural reservoirs for the orthoebolaviruses. After EBOV RNA was detected in three fruit bat species and the Egyptian rousette bat (ERB) was identified as the natural reservoir for Marburg virus (MARV; family Filoviridae, genus Orthomarburgvirus), the majority of research efforts in search for the natural reservoirs of the orthoebolaviruses have focused on fruit bats.

However, Angolan free-tailed bats (AFBs) were recently identified as a natural reservoir for Bombali virus (genus Orthoebolavirus) and circumstantially linked to the EBOV spillover event that initiated the 2014-2016 West African Ebola virus disease outbreak. To determine if AFBs could play a role in the maintenance and transmission of EBOV, Riesle-Sbarbaro et al. tested the permissiveness of AFBs for filovirus infection by experimentally inoculating captive AFBs with EBOV, MARV, Taï Forest virus (TAFV; genus Orthoebolavirus) and Reston virus (RESTV; genus Orthoebolavirus), and then testing oral, rectal, urine, and fecal samples, blood, and tissue samples for the presence of virus. The authors found that AFBs inoculated with EBOV exhibited high viral RNA loads in tissues, numerous including fetal tissue. Notably, the authors were able to isolate infectious EBOV from >10 tissue types (including fetal tissue), oral swabs, rectal swabs, and feces, indicating the potential for EBOV to be transmitted bat-to-bat by vertical and horizontal transmission routes. In contrast, AFBs inoculated with MARV, TAFV, and RESTV exhibited little to no virus replication.

This manuscript is very well written, easy to follow, and includes outstanding graphics. The study was comprehensive, conducted in a methodical manner, and is scientifically sound. Other than the specific comments listed below that include suggested typographical revisions or revisions to enhance clarity, I only recommend that the authors briefly discuss their opinion on what next steps should be taken given their results. For example, do the authors recommend intense, longitudinal sampling of wild AFBs to determine if they serve as a competent natural reservoir of EBOV in nature? Do the authors recommend similar experimental studies be performed with other insectivorous bat species to determine if more than one insectivorous bat species could be involved in EBOV maintenance in nature?

Specific Comments

Line 54: Filovirus taxonomy has been updated. Please see:
<https://link.springer.com/article/10.1007/s00705-023-05834-2>

Line 229: Suggest changing “Ag-positive” to “EBOV-positive”

Line 311: Insert “to be” after “thought”

Line 351: Suggest changing “persisted until” to “still present at 10 dpi” since no bats were euthanized after 10 dpi and the virus might persist for longer.

Line 353: Change “by” to “at”

Lines 384-386: High MARV loads detected in the oral mucosa of experimentally infected ERBs suggest that the virus can be transmitted bat-to-bat through biting.

Lines 411 and 412: Suggest providing an example of targeted preventative measures that could be implemented in the future.

Line 439: “Speciation” should be “species”. Speciation refers to the formation of distinct species in the course of evolution.

Line 493: Suggest including the actual inoculum dose here to avoid confusion (6×10^6 /mL/60 μ L = 6×10^4 TCID₅₀).

Lines 504-520: Please include procedures used to collect urine and feces.

Line 600: “asses” should “assess”.

Line 618: Insert a comma after “virus”.

Lines 671 and 675: Change “infected” to “infect”.

Line 679: Change “covert” to “convert”.

Line 679: Suggest inserting “nucleic acid” before “extracts”.

Table 1, lines 983-986: The text reports a PCR and isolation positive uterus for a MARV-inoculated AFB that is not shown in the table. The positive urine sample from an EBOV-inoculated AFB is not shown in the table.

Figure 1, lines 988-1009: The legend descriptions do not match all the figure panels.

Supplementary Table 1, Description of activity – pilot training, N5: Change “returned” to “return”.

Supplementary Table 2, lines 34-35: Should be “No statistically significant differences were detected”.

Reviewer #1 (Remarks to the Author):

Thank you for the opportunity to review this interesting and relevant paper. The authors describe experimental infection of an insectivorous bat species, the Angolan free-tailed bats and provide evidence that this species may be a potential reservoir of EBOV (previously Ebola Zaire). In addition, does the paper also provide important information on the husbandry of insectivorous bats in experimental infections that can inform future studies. The paper is well written, and I only have a few minor comments below;

General comments;

1. The authors should update the filovirus taxonomy, especially genus and species names and use of italics according to the recent publication; Biedenkopf et al., 2023, Archives of Virology (2023) 168:220, <https://doi.org/10.1007/s00705-023-05834-2> <<https://doi.org/10.1007/s00705-023-05834-2>>

We thank the reviewer for improving and updating our manuscript. We have now updated the vernacular and included the latest virus taxonomy:

Line 43: included (EBOV; *Orthoebolavirus zairense*)

Line 52: included (MARV; *Orthomarburgvirus marburgense*)

Line 55: included (RESTV; *Orthoebolavirus restonense*)

Line 56: changed “ebolavirus” to “orthoebolaviral”

Line 57-58: included (BOMV; *Orthoebolavirus bombaliense*)

Line 62, 169, 332, 386, 413: changed “ebolavirus” to “orthoebolavirus”

Line 81-82: included (TAFV; *Orthoebolavirus taiense*)

Line 123: changed “Ebolavirus” to “Orthoebolavirus”

Line 123: changed “Marburgvirus” to “Orthomarburgvirus”

Line 297: changed “marburgviruses” to “orthomarburgviruses”

Line 183, 189, 353, 383: changed “ebolaviruses” to “orthoebolaviruses”

Line 756, 785: changed “Zaire ebolavirus” to “Orthoebolavirus zairense”

Line 756-757: changed “Sudan ebolavirus” to “Orthoebolavirus sudanense”

Line 757: changed “Reston ebolavirus” to “Orthoebolavirus restonense”

Line 757-758: changed “Tai Forest ebolavirus” to “Orthoebolavirus taiense”

Line 758: changed “Bundibugyo ebolavirus” to “Orthoebolavirus bundibugyoense”

Line 759, 786: changed “Marburg marburgvirus” to “Orthomarburgvirus marburgense”

2. Ebolavirus has never been detected in the Angolan free-tailed bat, and this species has only been implicated through circumstantial evidence. Bombali virus on the other hand, has been detected. It would have added a lot of value to include Bombali virus in this study for comparative purposes. If not now, maybe in future work.

We thank the reviewer for the interest of improving our study and the suggestion. We completely agree with you! We are definitely aiming to evaluate the comparative infection kinetics and pathology, or lack of, of BOMV to those seen with other orthoebolaviruses in AFBs, particularly EBOV. After great effort, we are currently optimistic about resuming transportation of wild caught-bats and also of receiving WT-BOMV or rBOMV to our BSL4 facilities for future experiments.

3. Also linked to the point above. If the Angolan free-tailed bat is a potential reservoir for EBOV, why was it not detected in surveillance studies. Several surveillance efforts reported the presence of Bombali virus. I would have expected EBOV positives also if a reservoir. Can the authors please address this point in the discussion?

As you, we have evaluated this point. Although there have been increasing surveillance studies targeting molossid bats, the number of studies is still small. There are considerably more sero-surveillance efforts targeting fruit bats in comparison. For example, before 2018, less than 70 AFBs had been surveyed in 30 years of investigation of EBOV-reservoir hosts. After increasing the number of AFBs studied, BOMBV was discovered. However, additional to a lack of targeted surveillance, it is still uncertain how the dynamics of EBOV transmission compare to the ones of other filoviruses in bats. It could be possible that EBOV has a different strategy of maintenance, which can influence intermittent recrudescence in specific windows of a population's phenology.

We have now included:

Line 468: "...However, additional surveillance studies of, not only AFBs, but also other molossid populations are warranted to evidence EBOV natural infection in free-tailed bats. In fact, only after a large study in 2018, that surveyed 109 free-tailed bats including 52 AFBs, BOMV was discovered(1); before that year AFBs only played a minor role in orthoebolavirus surveys."

Specific comments;

Line 36-39 and Table 1; Virus isolation was not attempted for all PCR positive samples.

Can the authors indicate why, e.g. 10dpi for EBOV?

Indeed, because of the amount of PCR-positive samples, we evaluated a representative selection of the sample types, which were likely to be successfully titrated by TCID₅₀. To do so, we only assayed samples that had a minimum RNA copy numbers (now mentioned as RNA load cut-off) or that were of foetal origin and available for titration. Only 3 EBOV 10dpi samples were within this criterion: E05 lymph node (titration negative), E05 Placenta (titration positive) and E5o spleen (below the cut-off and titration negative). We have now detailed the criteria used to attempt virus isolation in our methods, section "virus isolation". In this section, we also specified the criteria used for samples that were below this RNA load cut-off, but were of importance to understand and evaluate potential transmission, both vertical and horizontal (i.e. foetal tissues, excretions and secretions).

We have included:

Line 571-572; "... tissues that had virus RNA copies/g above a cut-off of $>5 \times 10^5$ or Ct values <30 , or were of foetal origin and available for titration."

Line 584-585; "Selected samples that were below our cut-off: that had virus RNA copies/ml $<5 \times 10^5$ (or Ct values <30) but $>2 \times 10^2$ copies/ml, such as faeces, bladder and mucosal swab samples (oral and rectal), were homogenized..."

Also in:

Line 136-139 (we assumed that this specific comment referred to these lines, as lines 36-39 are within the abstract); we now included "Furthermore, from 16 representative sample-types (i.e. tissues with $>5 \times 10^5$ viral RNA loads, of foetal origin or secretions/excretions with $>2 \times 10^2$ RNA loads) infectious EBOV was isolated from 13 (Table 1), ..."

Table 1; we now included "(--) Virus isolation not attempted: virus RNA loads below the cut-off."

Line 292-294; Is it possible to indicate that BOMV also have a slow evolutionary clock based on very limited detections of this virus?

We thank the reviewer for this observation. Indeed, although there are BOMV sequences with great geographical distance, the time span of these is shorter and the amount of available sequences is far lower than what is available for MARV. Therefore we have now edited this sentence to "..., **is also suggested** for BOMV in populations ..."

Line 300; It indicates “original human isolates passages in Vero cells”. Is this not already adapted virus and how many times was it passaged?

Line 300 mentions “original human isolates passaged in Vero cells”. Although there is a difference between these two statements, this can be very nuance. Therefore we have now edited to “original human isolates passaged in, and potentially adapted to, Vero cells”. We can trace 4 passages of our EBOV isolate in Vero cell cultures. However, we do not have the same detailed traceability for MARV.

Line 327; I suggest changing: “not the filoviruses” to the specific ones included in this study. Not all filoviruses were included.

We have now changed “...filoviruses” to “... RESTV, TAFV and MARV”

Line 407-412; I find these last sentences very confusing. Why link the results to culling? Also, expand on the link with birthing seasons and specifically what targeted preventative measures can be used.

We appreciate the reviewer suggestion to improve the clarity of our manuscript. We have now edited this sentence to “We highlight the importance that a human-derived EBOV is capable to infect AFBs and readily transmit through vertical and potentially horizontal pathways. **We hypothesise that** birthing seasons could not only drive EBOV maintenance in reservoirs, but also spillover to humans. **However, blunt measures to stop virus transmission, such as culling, are not only detrimental to bat conservation but also have been shown to be counterproductive(2, 3). Instead, targeted preventative measures, such as seasonal closure of cave exploration activities (e.g. corresponding to AFB birthing seasons) and raising public awareness of likely seasonal risks to spillover, could be implemented in the future.**

Line 434; Add the country, Ivory Coast

We thank the reviewer for pointing this out. We have now added the country of origin in our Capture and Training methods section. Following our previous format, we included Côte d’Ivoire instead of Ivory Coast.

Line 441; Why only sequence such a short fragment? Also, add the GenBank accession number of these sequences for reference and indicate what analyses were used to confirm species identification. There have been several reclassifications in the *Mops* genus recently, and it needs to be clearly demonstrated that the species used is indeed *Mops condylurus*. I suggest adding this to the supplementary information to prove this point.

We agree with the reviewer that this fragment is rather short for species identification. During this study, and previous work within our research group, we have identified AFBs by sequencing this fragment, but also and firstly by morphological keys and characteristics that may help differentiate AFBs from other molossid species that occur in the area; to our knowledge there is no definitive evidence of cryptic species of molossid bats.

We have now included Supplementary Table 2 (we updated the suppl. table order), which shows a list of bats of the *Mops* genus (including the previously known *Chaerephon*) that co-occur in the region, describing the known morphological characteristics, e.g. average weight, forearm length, coat colour (dorsal and ventral) and specific keys (e.g. interaural crest presence, ventral hair strips and flanks). We also included the IUCN red list conservation status and the mitochondrial sequences availability in GenBank. Unfortunately, from 12 molossid bat species included in this table, only 2 currently have public sequences for *cyt b*: *Mops condylurus* and *Mops pumilus*, and 3 have partial sequences for COI: *Mops condylurus*, *Mops pumilus* and *Mops nanulus*.

Percent identity of the sequenced 241 bp *cyt b* fragment between bats of the species *M. condylurus* and *M. pumilus* differ by >10%, which would suffice for species identification. Nonetheless, these 2 species are easily distinguishable by key morphological features; the same can be said for *M. nanulus*.

We have not uploaded the short *cyt b* sequences to GenBank, as they share in average 99% identity to the published sequences of *M. condylurus* and 88.5% identity to the published sequences of *M. pumilus* (individual % identity included in Supplementary Table 2). We sequenced the animals for our own confirmation of species (after evaluation of the morphological features) rather than to be able to contribute significantly to this species taxonomy.

Line 464, Stage 2; Please comment on how many animals were transported and how many survived.

We have now included: "...a total of 33 animals **were shipped** and 32 arrived to our facility healthy; **one animal was unfit for the experiment, likely as a result of the extended captivity and air transportation: 7 males and 4 females from the capture in May (1 male euthanized after arrival);...**"

Line 505; Add more specific results on the testing of filovirus previous exposure. Was a full panel of filovirus antigens used for this purpose? Any testing for other virus infections?

Previous exposure to filoviruses was also investigated using the same validated Luminex assay that was used for post-infection serological evaluation (4). This methodology has been described in detail in our methods and result section and also has been cited. The assay recognizes Abs (validated for humans and bats) against EBOV and MARV –NP. The beads were coupled with full-length EBOV and MARV recombinant NP. The assay cross-reacts with other orthoebolaviruses and orthomarburgviruses, as shown in our experiment. However, unfortunately we haven't yet been able to test this assay with bat Abs raised against BOMV.

Before experimental infection, we evaluated previous exposure and potential active infection of lyssaviruses using a nested RT-PCR that can detect 7 genotypes (5), and to filoviruses using a RT-qPCR assay that has been validated to detect EBOV, SUDV, RESTV, TAFV, BDBV and MARV (6). Because the bats did not show signs of disease during the 4-week quarantine, we assume no active pathogenic virus infection; as AFBs are not known reservoir of other zoonotic viruses to our knowledge, we did not survey other viruses.

We have now included specific details about pre-experimental screening in the method section:

Line 593-598: "Oral swab samples were used to test exposure and active infection of lyssaviruses and filoviruses. Briefly, 7 genotypes within the *Lyssavirus* genus were screened using a nested RT-PCR(5); filovirus active infection was screened using a RT-qPCR assay that has been validated to detect EBOV, SUDV, RESTV, TAFV, BDBV and MARV(6). Previous exposure to filoviruses was screened using a previously validated Luminex assay(4)."

Table 1; page 30

We have now added a page break before the Table and Figures section

Reviewer #2 (Remarks to the Author):

The manuscript by Riesle-Sbarbaro et al. describes experimental filovirus infection of Angolan free-tailed bats. The manuscript is well-written and provides a highly-detailed experimental infection model with a broad analysis of the infectious process. The work strongly supports the hypothesis that this species of bat is a likely reservoir host of Ebola virus, a question that has remained enigmatic for a couple of decades. This manuscript is a significant step in understanding the ecology of Ebola virus.

The description for adapting the bats to feeding upon mealworms is excellent and may provide a guide for use of other insectivorous bats of importance to virus research.

This reviewer only has a few suggestions for the authors to consider. First, were any of the polymorphisms identified in figure 5 present in the stock virus preparation? Deep sequencing is unlikely to detect such rare variants, which could have been selected upon passage in cell culture or bats. Were multiple (e.g., 3) passages independently conducted to determine if the polymorphisms were identical? If the same polymorphisms persisted, it suggests they were present in a low number of viruses in the original stock. PCR with a primer spanning the polymorphism(s) could also address this question.

We thank the reviewer for this insight and the interest in our study. We indeed deep-sequenced our stock isolates (EBOV_VC and MARV_VC), which we then passaged twice in MoKi cells (inoculum for the *in vivo* infection). Our EBOV-stock had been passaged 4 times in Vero cells. Unfortunately, we do not have the same traceability for our MARV-stock isolate.

Several of the SNPs detected in our stock-isolate persisted in the later passages. We agree that the polymorphisms could have been selected in the later passaging. Our aim is to highlight the great difference of the selection between the viruses in AFBs or AFB-derived cells. Figure S9-S10 show variant calling from SNPs in reference to the mapping sequence, which include our stock-isolates. Because their figure legends were deficient in providing specific information, we have now included: “Percentages of each variant, in reference to the mapping sequence, ...” in both figures, also in Fig S9: “No variant calling represents the same base as the Map Seq and the 1% bar represents unknown base calling due to low coverage” and in Fig S10 “No variant calling represents the same base as the Map Seq”.

Does the qPCR assay discriminate between viral genomic and mRNA?

No, our qPCR assay detects only a short fragment and within the middle section of VP30 (no 3'/5' ends). Our qPCR results only show the targeted VP30.

With the suggestion that macrophages may be susceptible (SF6j) as any attempt made to generate macrophages from bat PBMC or bone marrow to determine if they were susceptible to infection?

Yes, there have been trials to grow macrophages from AFBs PBMC within our research group. We are also developing reagents specific to the bat species. This will increase the likelihood of a successful generation of macrophages for future *in vitro* susceptibility studies.

The authors should note in the discussion that a potential issue with using wild-caught bats is the possibility they may unknowingly harbor other infectious agents or be impacted by environmental factors that could influence the results and interpretations of the study.

We thank the reviewer for this suggestion. Indeed, investigations over the complex interactions of the virome/microbiome in wild animals are deeply underrepresented, particularly in bats. However, even though we were to know past exposures of these bats with other pathogenic infections, it would be hard to deeply understand the influence in the context of our results (e.g. sample size). In order to properly represent this idea, we would require a lengthy paragraph in the discussion section. Instead, we have included in the methods section the assays we used to test previous exposure to viruses. In the results section of the manuscript it is stated the origin of these bats, the homogenization of the microbiota as a result of the diet adaptation and the thorough histopathological analysis used to tease out the pathology that specifically co-localized with virus presence.

In the discussion, to further this point out we included:

line 333: “...we show that infection of **wild-caught** AFBs...”

line 348-350: “Although these wild-caught bats showed no signs of disease during the quarantine period, it is possible that other infectious agents could be interfering with our results.”

A few minor considerations:

We thank the reviewer for improving our manuscript and we have now changed all the mentioned suggestions.

Line 31. Changed to “Bombali virus” without italics.

L297. Changed to “specific variant switches host species.” (I believe this is the authors’ intent)

L392. Changed to “infectious pathogens can traverse”

L939. Changed “Since” (passage of time) to “Because”.

L401. Changed “utility” to “use”.

Reviewer #3 (Remarks to the Author):

Although Ebola virus (EBOV; family Filoviridae, genus *Orthoebolavirus*) was discovered almost 50 years ago, its natural reservoir has not been identified. Cumulative PCR, serological, and circumstantial epidemiological evidence indicates that bats are natural reservoirs for the orthoebolaviruses. After EBOV RNA was detected in three fruit bat species and the Egyptian rousette bat (ERB) was identified as the natural reservoir for Marburg virus (MARV; family Filoviridae, genus *Orthomarburgvirus*), the majority of research efforts in search for the natural reservoirs of the orthoebolaviruses have focused on fruit bats. However, Angolan free-tailed bats (AFBs) were recently identified as a natural reservoir for Bombali virus (genus *Orthoebolavirus*) and circumstantially linked to the EBOV spillover event that initiated the 2014-2016 West African Ebola virus disease outbreak. To determine if AFBs could play a role in the maintenance and transmission of EBOV, Riesle-Sbarbaro et al. tested the permissiveness of AFBs for filovirus infection by experimentally inoculating captive AFBs with EBOV, MARV, Tai Forest virus (TAFV; genus *Orthoebolavirus*) and Reston virus (RESTV; genus *Orthoebolavirus*), and then testing oral, rectal, urine, and fecal samples, blood, and tissue samples for the presence of virus. The authors found that AFBs inoculated with EBOV exhibited high viral RNA loads in tissues, numerous including fetal tissue. Notably, the authors were able to isolate infectious EBOV from >10 tissue types (including fetal tissue), oral swabs, rectal swabs, and feces, indicating the potential for EBOV to be transmitted bat-to-bat by vertical and horizontal transmission routes. In contrast, AFBs inoculated with MARV, TAFV, and RESTV exhibited little to no virus replication.

This manuscript is very well written, easy to follow, and includes outstanding graphics. The study was comprehensive, conducted in a methodical manner, and is scientifically sound. Other than the specific comments listed below that include suggested typographical revisions or revisions to enhance clarity, I only recommend that the authors briefly discuss their opinion on what next steps should be taken given their results. For example, do the authors recommend intense, longitudinal sampling of wild AFBs to determine if they serve as a competent natural reservoir of EBOV in nature? Do the authors recommend similar experimental studies be performed with other insectivorous bat species to determine if more than one insectivorous bat species could be involved in EBOV maintenance in nature?

Specific Comments

Line 54: Filovirus taxonomy has been updated. Please see:

<https://link.springer.com/article/10.1007/s00705-023-05834-2>

<<https://link.springer.com/article/10.1007/s00705-023-05834-2>>

We thank and appreciate the reviewer for proofing and updating our manuscript. We now changed all mentioned suggestions as well as updated the vernacular and included the latest virus taxonomy:

Line 43: included (EBOV; *Orthoebolavirus zairense*)
Line 52: included (MARV; *Orthomarburgvirus marburgense*)
Line 55: included (RESTV; *Orthoebolavirus restonense*)
Line 56: changed “ebolavirus” to “orthoebolaviral”
Line 57-58: included (BOMV; *Orthoebolavirus bombaliense*)
Line 62, 169, 332, 386, 413: changed “ebolavirus” to “orthoebolavirus”
Line 81-82: included (TAFV; *Orthoebolavirus taiense*)
Line 123: changed “Ebolavirus” to “Orthoebolavirus”
Line 123: changed “Marburgvirus” to “Orthomarburgvirus”
Line 297: changed “marburgviruses” to “orthomarburgviruses”
Line 183, 189, 353, 383: changed “ebolaviruses” to “orthoebolaviruses”
Line 756, 785: changed “Zaire ebolavirus” to “Orthoebolavirus zairense”
Line 756-757: changed “Sudan ebolavirus” to “Orthoebolavirus sudanense”
Line 757: changed “Reston ebolavirus” to “Orthoebolavirus restonense”
Line 757-758: changed “Tai Forest ebolavirus” to “Orthoebolavirus taiense”
Line 758: changed “Bundibugyo ebolavirus” to “Orthoebolavirus bundibugyoense”
Line 759, 786: changed “Marburg marburgvirus” to “Orthomarburgvirus marburgense”

Line 229: Changed “Ag-positive” to “EBOV-positive”

Line 311: Inserted “to be” after “thought”

Line 351: Changed “persisted until” to “still present at 10 dpi” since no bats were euthanized after 10 dpi and the virus might persist for longer.

Line 353: Changed “by” to “at”

Lines 384-386: High MARV loads detected in the oral mucosa of experimentally infected ERBs suggest that the virus can be transmitted bat-to-bat through biting.

We included: “... Including contact with urine, **high MARV loads detected in the oral mucosa of experimentally infected ERBs suggest that the virus can be transmitted bat-to-bat through biting.**”

Lines 411 and 412: Suggest providing an example of targeted preventative measures that could be implemented in the future.

We have now included: “targeted preventative measures, **such as seasonal closure of cave exploration activities (e.g. corresponding to AFB birthing seasons) and raising public awareness of likely seasonal risks to spillover**, could be implemented in the future.”

Line 439: “Speciation” changed to “species”. Speciation refers to the formation of distinct species in the course of evolution.

Line 493: Suggest including the actual inoculum dose here to avoid confusion (6×10^6 mL/60 μ L = 6×10^4 TCID₅₀).

We have now included: “total inoculum dose of **6×10^4 TCID₅₀ in a volume of 60 μ L...**”

Lines 504-520: Please include procedures used to collect urine and feces.

We have now included: “Group faeces and urine were collected daily. For this, faecal pellets were collected in empty 2 ml tubes and urine was collected using a dry polyester-tipped swab, stored in 500 μ l of PBS. The collected faecal pellets were weighted, diluted 1:10 in PBS and centrifuged for 10 minutes at 12000 x g for supernatant collection.”

Line 600: “asses” changed to “assess”.

Line 618: Inserted a comma after “virus”.

Lines 671 and 675: Changed “infected” to “infect”.

Line 679: Changed “covert” to “convert”.

Line 679: Inserted “nucleic acid” before “extracts”.

Table 1, lines 983-986: The text reports a PCR and isolation positive uterus for a MARV-inoculated AFB that is not shown in the table. The positive urine sample from an EBOV-inoculated AFB is not shown in the table.

We thank the reviewer for improving our manuscript.

We have now included the urine PCR positive sample and a uterus/placenta sample for both EBOV and MARV cohorts.

We also corrected Table 1 title to: “... PCR samples collected from **adult** filovirus-inoculated AFBs.”

Figure 1, lines 988-1009: The legend descriptions do not match all the figure panels.

We have now corrected:

We thank the reviewer for improving our manuscript. We have now corrected the figure descriptions to match the figure panels.

Supplementary Table 1, Description of activity – pilot training, N5: Change “returned” to “return”.

We have now changed “return” to “returned” in Supplementary Table 1 - pilot training, N5.

Supplementary Table 2, lines 34-35: Should be “No statistically significant differences were detected”.

We thank and appreciate the reviewer for the thorough revision of our manuscript. We have now edited the legend of Supplementary Figure 2.

References

1. T. Goldstein *et al.*, The discovery of Bombali virus adds further support for bats as hosts of ebolaviruses. *Nat Microbiol* **3**, 1084-1089 (2018).
2. M. Viana *et al.*, Effects of culling vampire bats on the spatial spread and spillover of rabies virus. *Science Advances* **9**, (2023).
3. B. R. Amman *et al.*, Marburgvirus Resurgence in Kitaka Mine Bat Population after Extermination Attempts, Uganda. *Emerging infectious diseases* **20**, 1761-1764 (2014).
4. R. Surtees *et al.*, Development of a multiplex microsphere immunoassay for the detection of antibodies against highly pathogenic viruses in human and animal serum samples. *PLoS neglected tropical diseases* **14**, e0008699 (2020).
5. S. Vázquez-Morón, A. Avellón, J. E. Echevarría, RT-PCR for detection of all seven genotypes of Lyssavirus genus. *Journal of virological methods* **135**, 281-287 (2006).
6. T. Rieger *et al.*, Evaluation of RealStar Reverse Transcription–Polymerase Chain Reaction Kits for Filovirus Detection in the Laboratory and Field. *Journal of Infectious Diseases* **214**, S243-S249 (2016).

REVIEWERS' COMMENTS

Reviewer #1 (Remarks to the Author):

Thank you for addressing all my comments in detail and adding additional information. I have no more additional comments.

Reviewer #2 (Remarks to the Author):

The authors have sufficiently addressed my comments. This is an important work and should be published as soon as possible.